



# A triple tree-ring constraint for tree growth and physiology in a global land surface model

Jonathan Barichivich[1,2], Philippe Peylin[1], Thomas Launois[1], Valerie Daux[1], Camille Risi[3], Jina Jeong[4], and Sebastiaan Luyssaert[4]

[1]Laboratoire des Sciences du Climat et de l'Environnement (LSCE), Gif sur Yvette, France.
[2]Instituto de Conservación Biodiversidad y Territorio, Universidad Austral de Chile, Valdivia, Chile.
[3]Laboratoire de Météorologie Dynamique (LMD), Paris, France.
[4]Department of Ecological Sciences, VU University, 1081HV Amsterdam, The Netherlands.

**Correspondence:** Jonathan Barichivich (jonathan.barichivich@ipsl.lsce.fr)

**Abstract.** Annually-resolved tree-ring records extending back to pre-industrial conditions have the potential to constrain the responses of global land surface models at interannual to centennial time scales. Here, we demonstrate a framework to constrain the representation of tree growth and physiology in the ORCHIDEE global land surface model using the simulated variability of tree-ring width and carbon ($\Delta^{13}$C) and oxygen ($\delta^{18}$O) stable isotopes in six sites in boreal and temperate Europe. We exploit
the tree-ring triplet to derive integrative constraints for leaf physiology and growth from well-known mechanistic relationships among the variables. The model simulates $\Delta^{13}$C ($r = 0.31\text{-}0.80$) and $\delta^{18}$O ($r = 0.36\text{-}0.74$) better than tree-ring width ($r < 0.55$), with an overall skill similar to that of other models. The results show that growth variability is not well represented, and that the parameterization of leaf-level physiological responses to drought stress in the temperate region can be improved with tree-ring data. The representation of carbon storage and remobilization dynamics is critical to improve the realism of
simulated growth variability, temporal carrying over and recovery of forest ecosystems after climate extremes. The simulated physiological response to rising $CO_2$ over the 20th century is consistent with tree-ring data in the temperate region, despite an overestimation of seasonal drought stress and stomatal control on photosynthesis. Photosynthesis correlates directly with isotopic variability, but the origin of correlations with $\delta^{18}$O is not entirely physiological. The integration of tree-ring data and land surface models as demonstrated here should contribute towards reducing current uncertainties in forest carbon and water
cycling.

## 1 Introduction

A major challenge for the Land Surface Model (LSM) component of the Earth System models currently used to project climate change is to accurately simulate the historical and future dynamical coupling between the global biosphere and climate (Friedlingstein et al., 2014). Although LSMs are skillful at reproducing short-term (<20 years) contemporary observations
of plant water and carbon cycling, their responses at longer time scales from decades to century are still highly uncertain and contribute to the spread in current climate change projections (Ciais et al., 2013; Friedlingstein et al., 2014). Some of these models project that the terrestrial biosphere will continue behaving as a carbon sink of anthropogenic emissions during





the course of the century, while others simulate that it will turn into an additional carbon source to the atmosphere that will accelerate climate change (Friedlingstein et al., 2006; Jones et al., 2013; Friedlingstein et al., 2014). The uncertainties in simulated long-term trends are also evident for the water cycle and over the historical period (Phillips et al., 2019). For instance, global models simulate trends in transpiration of European forests during the 20th century ranging from -7 to 9% for evergreen

conifers and 3 to 26% for deciduous broadleaf forests (Frank et al., 2015). The lack of a general agreement on the historical and future long-term responses of the terrestrial biosphere in land surface models limit confidence in future climate projections (Ciais et al., 2013).

The development, parameterization and evaluation of current land surface models have been based on a handful of manipulative experiments (Ainsworth and Long, 2005; Smith et al., 2015; Andresen et al., 2016; Song et al., 2019), a quasi-global

network of eddy-covariance observations, Earth observations and forest inventories. Most of these data streams are not able to reveal the temporal evolution of plant responses to global change factors at multi-decadal and longer time scales, where mechanistic understanding on how trees adapt or perish to environmental changes is still limited (McDowell et al., 2008; Cailleret et al., 2018). Empirical tree-ring studies are increasingly being used to address the lack of direct observations on long-term changes in plant physiology and growth with global change (Huang et al., 2007; Frank et al., 2015; Babst et al., 2018; Zuidema

et al., 2018; de Boer et al., 2019).

The width, anatomy and chemistry of tree rings are unique integrators of responses to climatic and micro-climatic conditions directly affecting photosynthetic rates, carbon allocation patterns and cambial growth dynamics of trees (Fritts, 1976). Commonly measured tree-ring parameters such as ring width, density, anatomy and carbon and oxygen stable isotopes are therefore ideally suited to study the range of responses of mature trees to global change (Gunderson and Wullschleger, 1994;

Norby et al., 1999; Kirdyanov et al., 2008; Battipaglia et al., 2013), especially if used together and for multiple species across a gradient of functional traits or environmental conditions (McCarroll et al., 2003; Peñuelas et al., 2011; Cernusak and English, 2015; Hartl-Meier et al., 2015; Levesque et al., 2019). However, most of currently available tree-ring data have been collected for climate reconstruction purposes rather than for estimating forest growth changes (Klesse et al., 2018; Zhao et al., 2019). Re-purposing this vast archive to estimate and upscale growth trends from tree-level ring width data is still a major challenge

in the tree-ring community (Bowman et al., 2013; Brienen et al., 2017; Babst et al., 2018). Most of earlier attempts have been cursed by sampling biases associated with the preferential collection of the largest and oldest trees in dendroclimatology (Cherubini et al., 1998; Brienen et al., 2012; Nehrbass-Ahles et al., 2014; Duchesne et al., 2019). Yet, annual tree-rings are the only practical method of assessing historical growth and physiological responses over the lifespan of long-lived organisms like trees.

In recent decades, some dendrochronological and forest process-based models have successfully integrated the simulation of tree-ring width with carbon or oxygen isotopic ratios in order to interpret measurements and improve simulated tree water status, photosynthesis and growth (Fritts et al., 1999; Hemming et al., 2001; Ogée et al., 2009; Eglin et al., 2010; Danis et al., 2012; Wei et al., 2014; Ulrich et al., 2019). These models typically simulate short-term radial growth with different levels of complexity from the daily dynamics of cambial cells and wood formation (i.e., cell division, enlargement, and wall thickening) to the carbon and water balance of an individual tree. Because the simulation of cell dynamics becomes complex





and computationally expensive for scales larger than an individual tree, most forest models simulate radial growth based solely on carbon allocation to stem (Deleuze et al., 2004; Misson, 2004; Sato et al., 2007; Danis et al., 2012; Li et al., 2014).

A dendrochronological model of intermediate complexity is MAIDENiso (Danis et al., 2012). It simulates annual growth
increment at the tree level based on daily carbon allocation to leaves, stem, roots and storage as a function of climate, soil water balance and atmospheric $CO_2$. Along annual tree-ring width, it concurrently simulates daily oxygen and carbon isotopic ratios. This triple tree-ring capability makes MAIDENiso a good reference for benchmarking mechanistic developments in larger scale vegetation models.

Tree rings have only recently been considered for the evaluation of models in the global land surface modelling community.
However, currently there is no global land surface model with the capability to explicitly simulate tree-ring width, despite the variety of approaches available to describe radial growth at different temporal and spatial scales. Nevertheless, tree-ring width data have been used as an indirect benchmark for the interannual variability of simulated Net Primary Productivity in some global land surface models (Rammig et al., 2014, 2015; Churakova et al., 2016). In contrast to ring width, tree-ring carbon ($\delta^{13}$C) or oxygen ($\delta^{18}$O) stable isotopes have already been incorporated in some global land surface models such as FOREST-
BGC (Panek and Waring, 1997), ORCHIDEE (Shi et al., 2011; Churakova et al., 2016; Risi et al., 2016), JULES (Bodin et al., 2013), LPX-Bern (Saurer et al., 2014; Keel et al., 2016; Keller et al., 2017) and CLM4.5 (Raczka et al., 2016; Keller et al., 2017).

The comparison of simulated $\delta^{13}$C and derived physiological indicators such as the carbon isotopic discrimination ($\Delta^{13}$C) by plants and the intrinsic water use efficiency (iWUE) with direct tree-ring isotopic measurements has helped benchmarking
stomatal responses to drought stress and rising atmospheric $CO_2$ concentrations in the global models (Panek and Waring, 1997; Bodin et al., 2013; Saurer et al., 2014; Keller et al., 2017). Simulated $\delta^{18}$O has been used to evaluate the representation of hydrological processes along the soil-plant-atmosphere continuum (Risi et al., 2016) and interpret the variability in tree-ring $\delta^{18}$O data in terms of climatic drivers and source water $\delta^{18}$O (Shi et al., 2011; Keel et al., 2016; Churakova et al., 2016). The interannual variability of tree-ring $\delta^{13}$C and $\delta^{18}$O has been shown to correlate with local eddy-covariance measurements of
forest productivity (Belmecheri et al., 2014; Tei et al., 2019). A recent study showed that the relationship holds at regional scale and is stronger for $\delta^{18}$O than $\delta^{13}$C and tree-ring width (Levesque et al., 2019), suggesting that tree-ring isotopic variability might integrate large-scale physiological signals useful to evaluate global carbon cycle models. Yet, stable isotope-productivity relationships have not been studied in this type of models.

Empirical tree-ring studies typically combine $\delta^{18}$O and $\delta^{13}$C as a means to mechanistically interpret plant physiological
responses (Saurer et al., 1997; Scheidegger et al., 2000; Barnard et al., 2012; Roden and Farquhar, 2012). In contrast, most of the studies with land surface models have focused on the simulation and evaluation of a single tree-ring variable (Saurer et al., 2014; Keel et al., 2016; Keller et al., 2017). Key developments in the ORCHIDEE global land surface model now allow to explicitly simulate radial growth (Bellassen et al., 2010; Naudts et al., 2015; Jeong et al., 2020) and carbon and oxygen composition in tree-ring cellulose (Risi et al., 2016). This offers the opportunity to explore how multiple tree-ring variables can
be used to constrain the long-term plant responses simulated by global land surface models and identify processes that need to be better represented or parameterized in these models.





The aims of this study are to (i) integrate key developments and identify the critical processes to concurrently simulate the interannual variability of tree-ring width and its carbon ($\Delta^{13}$C) and oxygen ($\delta^{18}$O) stable isotopes in the ORCHIDEE global land surface model, (ii) develop a conceptual triple tree-ring constraint for simulated growth and physiology that exploits the mechanistic relationships among tree-ring variables, and (iii) evaluate the simulated relationships between productivity and tree-ring carbon and oxygen stable isotopes. We first assess and compare the ability of ORCHIDEE and the MAIDENiso tree-ring model (Danis et al., 2012) to simulate 20th century tree-ring width and isotopic variability, and their inter-relationships, climate responses and relationships with productivity in the Fontainebleau forest in France. Then, we run ORCHIDEE in five other sites along a climate gradient from boreal Finland to temperate France, and compare its ability to simulate $\delta^{18}$O variability in the gradient with that of the LPX-Bern global vegetation model (Keel et al., 2016).

## 2 Model and evaluation data

### 2.1 Model description

The global land surface model ORCHIDEE (Krinner et al., 2005) is the terrestrial component of the IPSL (Institut Pierre Simon Laplace) Earth System model (Dufresne et al., 2013; Boucher et al., 2020). ORCHIDEE simulates the half-hourly exchange of energy, carbon and water between the terrestrial biosphere and the atmosphere, either coupled with the LMDz (Laboratoire de Météorologie Dynamique Zoom) general circulation model (Hourdin et al., 2006) or forced by observed meteorology.

The global forests are represented by eight Plant Functional Types (PFTs) described by a common set of governing equations with specific parameter values, with the only exception of some PFT-specific phenology representations (Krinner et al., 2005). Canopy photosynthesis is based on the leaf-level photosynthesis formulation of Farquhar et al. (1980) and together with respiration, the energy balance and hydrological processes are simulated at a half-hourly time step, which is the typical resolution of eddy-covariance measurements of carbon and water fluxes (Baldocchi et al., 2001). Leaf gas exchange is simulated by coupling the photosynthesis model with the stomatal conductance model of Ball et al. (1987). The $CO_2$ demand is determined as the minimum of Rubisco carboxylation and RuBP regeneration, while $CO_2$ supply depends on the difference in $CO_2$ concentration between the air outside the leaf and the carboxylation sites. Carbon allocation to the different vegetation pools, phenology and mortality are calculated at a daily time step.

Soil hydrology is modelled using a surface and a deep reservoir (Choisnel et al., 1995) instead of the more complex multilayer soil diffusion scheme introduced in later versions of ORCHIDEE (Guimberteau et al., 2014). Water enters the surface layer via throughfall, snow melt, and dew and frost. Water can leave the soil reservoir through transpiration, bare soil evaporation, surface runoff and drainage. Plant water stress is calculated at half-hourly time steps as a function of soil water content (McMurtrie et al., 1990) weighted by root mass. Water stress reduces transpiration through a direct reduction of stomatal conductance as soil moisture depletes.

The key developments used for simulating tree-ring parameters include the addition of a forest management module (Bellassen et al., 2010) and the simulation of oxygen stable isotope ratios along the soil-plant-atmosphere continuum (Risi et al., 2016). The model code used in this study (SVN r898 version) precede the current code that merged the nitrogen cycle and





canopy structure in ORCHIDEE r5698 (Naudts et al., 2015; Vuichard et al., 2019), where the tree-ring functionality will also be ported (Jeong et al., 2020).

### 2.1.1 Tree-ring width

The forest management module explicitly simulates the temporal trajectory of stem growth of trees in a size-structured forest stand. The size structure of the stand is prescribed by defining a starting density of trees per hectare distributed across a number of stem diameter size classes, typically following an inverted-J distribution. Initial tree height and biomass are obtained by allometric relationships with stem diameter (Bellassen et al., 2010). For each PFT, photosynthesis and NPP are computed at the stand level at half-hourly and daily steps, respectively.

At the end of the year, accumulated woody NPP increment is allocated across the different diameter classes following an empirical competition rule for even-aged stands (Deleuze et al., 2004). A greater NPP share is allocated to larger dominant trees compared to smaller less vigorous trees, emulating the competition for light and resources within the stand. The absolute annual stem growth increment in the allocation rule is determined by a parameter ($\lambda$) that defines the slope between the fraction of NPP allocated to each diameter class and the mean diameter of the class. A second parameter ($\sigma$) represents a diameter threshold

below which less vigorous trees receive only a fixed part of the yearly stand NPP increment. These parameters are saved through the simulation and allow reconstructing the growth trajectory of each diameter class. In addition, the stem growth of the largest and smallest tree is tracked individually through the entire simulation (Bellassen et al., 2010).

   Simulated tree-ring width (i.e., radial growth) for each diameter class was computed from annual increments in stem circumference as follows:

$$\text{TRW}_t = \frac{Circ_t - Circ_{t-1}}{2\pi} \qquad (1)$$

where $\text{TRW}_t$ is the annual tree-ring width in millimetres for year $t$ and $Circ_t$ and $Circ_{t-1}$ are the stem circumferences for the current year and the previous year, respectively. The annual ring-widths simulated in this scheme account for the size-related trend in stem growth increment and PFT-specific tree allometry, providing a mean representation of radial growth for trees across a range of sizes that can be meaningfully compared with measured tree-ring width data. However, in this formulation

tree-ring width variability still depends only on direct GPP allocation (i.e., it is a carbon source driven process) and does not consider the dynamics of carbon storage or wood formation processes that can account for a large fraction of observed interannual tree-ring width variation (Misson, 2004).

### 2.1.2 Tree-ring carbon isotopes

To distinguish between variations in the $\delta^{13}$C of atmospheric $CO_2$ and the effects of plant metabolic processes, the $^{13}$C enrich-

ment of plant organic material is usually expressed in terms of the carbon isotope discrimination ($\Delta^{13}$C). Differences between the $^{13}$C enrichment of atmospheric $CO_2$ and plant material are attributed to discriminatory processes during photosynthesis.





Photosynthetic carbon isotope discrimination was estimated at half-hourly time steps using the simple formulation of Far-quhar et al. (1982) for C3 plants:

$$\Delta^{13}\text{C} = a + (b-a)\frac{c_i}{c_a} \tag{2}$$

where $a$ (4.4‰) is the kinetic discrimination associated with diffusion between free air and the stomatal cavity, $b$ (27‰) is the
fractionation during $CO_2$ fixation by the Rubisco enzyme, $c_i$ is the leaf internal $CO_2$ concentration simulated by ORCHIDEE
and $c_a$ is the atmospheric $CO_2$ concentration prescribed from measurements. Annual tree-ring carbon discrimination ($\Delta^{13}\text{C}_\text{R}$)
was then calculated as the mean of the half-hourly discrimination values in the above-ground biomass weighted by Gross
Primary Productivity (GPP) between the start ($SOS$) and the end ($EOS$) of the growing season:

$$\Delta^{13}\text{C}_\text{R} = \frac{1}{\sum_{n=SOS}^{EOS}\text{GPP}(n)}\sum_{n=SOS}^{EOS}\left(\Delta^{13}\text{C}(n) \times \text{GPP}(n)\right) \tag{3}$$

This formulation for carbon discrimination is commonly used as a simple approximation for discrimination derived from
measured $\delta^{13}\text{C}$ in tree-ring cellulose (Francey and Farquhar, 1982). For simplicity, it assumes that further post-photosynthetic
fractionation during photo- and dark-respiration and carbohydrate remobilization and storage is negligible. Although these
processes normally have an important impact on whole-ring cellulose isotopic composition (Gessler et al., 2009; Werner et al.,
2012), at least the impact of carbon remobilization is minimal in the latewood component of tree rings (Helle and Schleser,
2004). The $c_i$ term in Eq. (2) integrates the gas exchange dynamics (i.e., stomatal conductance and photosynthesis) simulated
by the model as a complex function of micrometeorological variability, seasonal water stress, and the long-term warming and
increase in $c_a$. Hence, despite its simplicity, simulated carbon discrimination using Eq. (2) has been shown to be valuable to
constrain the integrated environmental response of land surface models using carbon discrimination derived from tree-ring
$\delta^{13}\text{C}$ measurements (e.g., Bodin et al., 2013; Churakova et al., 2016; Keel et al., 2016).

### 2.1.3 Tree-ring oxygen isotopes

The $\delta^{18}\text{O}$ signature in tree-ring cellulose reflects primarily the isotopic composition of soil water and the evaporative enrich-
ment of leaf water due to transpiration, but other mixing and biochemical fractionation processes during water transport along
the soil-plant-atmosphere continuum also contribute to the final $\delta^{18}\text{O}$ signature (McCarroll and Loader, 2004; Gessler et al.,
2014). The $\delta^{18}\text{O}$ isotopic composition of the source water used by plants may originate from rainfall or groundwater, while the
$\delta^{18}\text{O}$ isotopic enrichment of leaf water depends strongly on vapour pressure deficit (i.e., the ratio of the vapour pressure in the
atmosphere and the intercellular spaces of the leaves) or relative humidity (Farquhar et al., 1998; Scheidegger et al., 2000).

The $\delta^{18}\text{O}$ fractionation and mixing processes in all water pools and fluxes along the soil-plant-atmosphere continuum (Risi
et al., 2016) is represented following a similar formulation than in other isotope-enabled global land surface models (Aleinov
and Schmidt, 2006; Haese et al., 2012). The isotopic compositions of precipitation and near-surface water vapour have to be
prescribed monthly when running the model stand-alone or are simulated by the LMDZ general circulation model in coupled
simulations. Precipitation reaching the soil surface or intercepting the canopy fractionates during evaporation according to the





Craig and Gordon equation (Craig and Gordon, 1965), which generically describes the preferential evaporation of the lighter isotope of a free water body at steady state.

The resulting isotopic composition of soil water and mixing are parameterized as a vertical profile to overcome the limitation

of depth resolution in the two-layer representation of soils in the model. No isotopic fractionation is assumed to occur during absorption of soil water by roots and thus the isotopic signature of xylem water is the same as that of soil water. The isotopic composition of leaf water at the evaporation sites ($\delta^{18}O_e$) is diagnosed by inverting the Craig and Gordon (1965) equation:

$$\delta^{18}O_e = \alpha\left(\alpha_k \times (1-h) \times \delta^{18}O_{sw} + h \times \delta^{18}O_v\right) \tag{4}$$

where $\delta^{18}O_{sw}$ is the isotopic composition of soil water taken up by the roots integrating older soil water and recent precipita-

tion, $\delta^{18}O_v$ is the isotopic composition of atmospheric water vapour, $\alpha$ is the equilibrium fractionation due to the phase change from liquid water to vapour, $\alpha_k$ is the kinetic fractionation due to diffusion of vapour into unsaturated air, and $h$ is the relative humidity normalized to surface temperature.

Isotopic enrichment of leaf water in the mesophyll ($\delta^{18}O_{lw}$) results from mixing between isotopically enriched leaf water at the evaporative site and depleted xylem water ($\delta^{18}O_{xw}$) through the so-called Peclet effect:

$$\delta^{18}O_{lw} = \delta^{18}O_e \times f + \delta^{18}O_{xw} \times (1-f) \tag{5}$$

where $f = (1 - e^{-P})/P$ is a coefficient that decreases as the Peclet effect increases, with $P = (E \times L)/(W \times D)$ as the Peclet parameter. $E$ is the transpiration rate per leaf area, $L$ is the effective diffusion length and $W$ is the leaf water content per leaf volume. $L$ was set to 8 mm and $W$ was assumed to be 103 kg m$^{-3}$ (Risi et al., 2016).

Half-hourly tree-ring cellulose isotopic composition ($\delta^{18}O_{cell}$) is calculated from the isotopic composition of leaf water

($\delta^{18}O_{lw}$) and xylem water ($\delta^{18}O_{xw}$) following the formulation of Anderson et al. (2002):

$$\delta^{18}O_{cell} = (f0 \times \delta^{18}O_{xw} + (1-f0) \times \delta^{18}O_{lw}) \times (1+\epsilon) \tag{6}$$

Where $f0$ is the fraction of leaf water exchanged with xylem water prior to cellulose synthesis, which reduces the imprint of leaf water on cellulose. For $\delta^{18}O$ this exchange is estimated to be 0.42 based on the best fit relationship under controlled experiments (Roden et al., 2000). The parameter $\epsilon$ is the biochemical fractionation factor during cellulose formation associated

with water carbonyl interactions and is estimated to be 27‰ (DeNiro and Epstein, 1979; Stenberg and DeNiro, 1983).

An estimate of growing season tree-ring isotopic composition ($\delta^{18}O_R$) is obtained by weighting $\delta^{18}O_{cell}$ by GPP as done for $\Delta^{13}C_R$ in Eq. (3):

$$\delta^{18}O_R = \frac{1}{\sum_{n=SOS}^{EOS} GPP(n)} \sum_{n=SOS}^{EOS} \left(\delta^{18}O_{cell}(n) \times GPP(n)\right) \tag{7}$$

A model evaluation across a network of 10 sites in Europe and North America shows that this representation is able to reproduce the main features of the seasonal and vertical variations in soil water isotope content, as well as the seasonal isotopic





signals in stem and leaf water (Risi et al., 2016). The isotopic variability simulated by ORCHIDEE has been used to interpret local climatic signals in boreal tree-ring $\delta^{18}$O records (Shi et al., 2011; Churakova et al., 2016) and to investigate regional and global isotopic signatures of continental recycling (Risi et al., 2013).

**Table 1.** Location and characteristics of the six tree-ring sites used for model evaluation. The ORCHIDEE Plant Functional Types (PFTs) corresponding to the sites are Boreal Needleleaf Evergreen (BoNE), Boreal Broadleaf Summergreen (BoBS) and Temperate Broadleaf Summergreen (TeBS).

| | Site | | | | | |
|---|---|---|---|---|---|---|
| | Kessi | Sivakkovaara | Bromarv | Rennes | Fontainebleau | Annecy |
| Country | Finland | Finland | Finland | France | France | France |
| Species | Pine | Pine | Oak | Oak | Oak | Oak |
| ORCHIDEE PFT | BoNE | BoNE | BoBS | TeBS | TeBS | TeBS |
| Latitude | 68.6 | 62.5 | 60.0 | 48.1 | 48.4 | 45.9 |
| Longitude | 28.2 | 31.2 | 23.0 | -1.7 | 2.7 | 6.2 |
| Elevation (m) | 159 | 200 | 5 | 70 | 100 | 450 |
| Stand age (years) | 400 | 400 | 150 | 120 | 120 | 70 |
| Stand density (trees ha$^{-1}$) | - | - | - | 240 | 140 | 40 |
| No. trees for tree-ring width | 16 | 28 | 7 | 28 | 15 | 13 |
| No. trees for tree-ring isotopes | 4 | 4 | 4 | 4 | 4 | 9 |
| Soil type | Sandy loam | Sandy loam | Sandy loam | Loam | Calcareous loam | Clay loam |
| Soil depth (m) | <0.5 | <0.5 | 0.5 | 1.5 | 1.0 | - |
| Annual precipitation (mm) | 390 | 582 | 502 | 677 | 678 | 906 |
| Annual mean minimum temperature ($^o$C) | -4.8 | -1.8 | 4.5 | 7.6 | 6.4 | 6.6 |
| Annual mean maximum temperature ($^o$C) | 4.3 | 5.8 | 9.1 | 16.0 | 15.4 | 16.2 |

## 2.2 Tree-ring data

Six previously published tree-ring sites in northern and western Europe with simultaneous measurements of ring-width and $\delta^{13}$C and $\delta^{18}$O compositions were used to evaluate the skill of the model along a climate gradient (Table 1). The three northernmost sites are located in the temperature-limited boreal region in Finland, and have chronologies of ring-width and $\delta^{13}$C and $\delta^{18}$O composition for Scotch pine (Kessi and Sivakkovaara; Hilasvuori et al., 2009) and Pedunculate oak (Bromarv; Hilasvuori and Berninger, 2010). The remaining three sites are located in France and represent moisture-sensitive temperate forests of Sessile oak (Fontainebleau; Etien et al., 2009) and Pedunculate oak (Rennes and Annecy; Raffalli-Delerce et al.,





2004; Danis et al., 2006). In all sites, except Annecy, the available ring-width and $\delta^{13}$C and $\delta^{18}$O chronologies cover the common period 1960–2000.

The tree-ring width measurement series available for each site were standardized by dividing each series by its mean ring
width (Cook et al., 1990). The resulting series of tree-ring width indices were averaged together to produce a mean site chronology composed by seven to twenty-eight trees (Table 1). This simple standardization method allows the computation of average tree-ring chronologies without the average being dominated by the faster growing trees with wider rings. Depending on the site, 4 to 9 trees were selected to develop carbon ( $\delta^{13}$C$_R$) and oxygen ($\delta^{18}$O$_R$) stable isotope chronologies for each site by pooling rings from the selected trees (Table 1) using standard methods for cellulose extraction and measurement of the
isotopic ratios (McCarroll and Loader, 2004; Daux et al., 2018). For oaks, earlywood and latewood sections were separated but only latewood was processed. For pine, the carbon and oxygen isotopic compositions were measured for the whole ring.

Tree-ring carbon discrimination was computed by subtracting the stable carbon isotope composition of the atmosphere ($\delta^{13}$C$_a$) from the measurements using the expression of Farquhar et al. (1982):

$$\Delta^{13}\text{C}_R = \frac{\delta^{13}\text{C}_a - \delta^{13}\text{C}_R}{1 + \frac{\delta^{13}\text{C}_R}{1000}} \tag{8}$$

For Fontainebleau, the intrinsic Water Use Efficiency (iWUE) was calculated from $c_i$ and $c_a$ for the period 1901–2000 following the expression (Farquhar et al., 1982):

$$\text{iWUE} = c_a(1 - \frac{c_i}{c_a}) \times 0.625 \tag{9}$$

$c_i$ was computed by reordering the terms of Eq. (2). Data for $\delta^{13}$C$_a$ and $c_a$ were obtained from McCarroll and Loader (2004) for the period prior to 1997 and from the Scripps website (http://scrippsco2.ucsd.edu/) from 1998 onwards. The three
well-known theoretical scenarios for gas exchange under rising $CO_2$ (constant $c_i$, constant $c_i/c_a$ and constant $c_a$-$c_i$) of Saurer et al. (2004) were used as a guideline to interpret the observed and simulated changes in iWUE over the 20th century. These scenarios differ only in the degree in which the increase in $c_i$ follows the increase in $c_a$ (either not at all, in a proportional way, or at the same rate, respectively). Mean $c_i$ over 1901–1910 was used as the starting point for the scenarios.

### 2.3 Simulations

The model was run at each tree-ring site over the period 1901–2000 using as meteorological forcing the nearest $0.5^o$ grid of the 6-hourly CRU-NCEP dataset (Viovy, 2014). This gridded forcing dataset is a statistical merging of the monthly CRU TS station-based dataset of the Climate Research Unit (New et al., 2000) with the atmospheric reanalysis from the National Center for Environmental Prediction (NCEP). The corresponding soil type and vegetation PFT were prescribed for the sites (Table 1). Monthly $\delta^{18}$O composition of precipitation and water vapour were obtained from the nearest grid of a global simulation of the
isotope-enabled LMDz general circulation model nudged by an atmospheric reanalysis over the period 1890-2007 (Risi et al., 2010).

Biomass and soil carbon pools were initialized to steady state equilibrium by a 5000-year spinup obtained by cycling over the meteorology for the period 1901–1910. The model was then run over the period 1901–2000 using observed $CO_2$ and an





initial tree density of 1000 trees per hectare to approach current forest age and density following tree mortality over time due to self-thinning.

For a comparison with ORCHIDEE, a simulation of the MAIDENiso tree-ring model for the Fontainebleau forest was obtained from an earlier study (Danis et al., 2012). The simulation was produced using a site-specific calibration and includes tree-ring width, $\Delta^{13}$C and $\delta^{18}$O over the period 1953–2000. In addition, simulated tree-ring $\delta^{18}$O with the LPX-Bern Dynamic Global Vegetation Model was extracted for each site from a published global simulation available for the period 1960–2012 at $3.75^o$x$2.75^o$ (Keel et al., 2016, dataset available at www.climate.unibe.ch).

## 2.4  Model-data comparison

The ability of ORCHIDEE to simulate the interannual variability of the three tree-ring parameters over the course of the 20th century was first evaluated in the Fontainebleau forest (Table 1). Fontainebleau is a well-studied tree-ring site in France (Michelot et al., 2011, 2012; Daux et al., 2018) and it has been used to evaluate the MAIDENiso tree-ring model (Danis et al., 2012). We compared our simulation of ring-width, $\Delta^{13}$C and $\delta^{18}$O for this site with the simulation of MAIDENiso over the period 1953–2000. Then, the evaluation was conducted in the rest of the sites over the common period 1960–2000, except in Annecy where a shorter span of the observations limited the model-data comparison to the period 1971–2000.

A simulated tree-ring width chronology was derived for each site by dividing the simulated tree-ring width series of the largest model tree by its mean. Since growth allocation in the model increases almost linearly with stem size, the absolute annual ring-width vary among stem size-classes but its interannual variability remains similar across all size classes, thus the choice of size class does not affect the standardized variability. The standardization removes the effect of stem size-class and conserves the interannual and longer variability but does not remove the juvenile effect in tree-ring width. However, the juvenile trend in simulated and observed ring-width does not affect the evaluation period (1960-2000) because at this time trees were already mature canopy individuals with their radial growth fluctuating around the mean.

Simulated $\delta^{18}$O and $\Delta^{13}$C tree-ring chronologies were obtained by averaging the simulated half-hourly isotopic variability between May and August. Using just the summer season (Jun–August) improves results for $\Delta^{13}$C but degrades substantially results for $\delta^{18}$O, thus May was included as a compromise in order to use a common season for the isotopes and ensure comparability across sites. A mean $\delta^{18}$O series was created from the nearest LPX-Bern grid ($3.75^o$x$2.75^o$) and corresponding PFT to each evaluation site for a comparison with the observations and ORCHIDEE over the common period 1960–2000.

Since the focus of our evaluation is on the interannual variability and not on the absolute values, correlation and the normalised standard deviation (i.e., the standard deviation of simulated tree-ring parameters divided by the standard deviation of the observations) were used to quantitatively evaluate the skill of the models to simulate the variability in tree-ring width and stable isotopes. Differences in the temporal persistence (i.e., carrying over effect) in the observed and simulated tree-ring parameters were evaluated using the first-order autocorrelation of the time series. The climate response of simulated ring width, $\Delta^{13}$C and $\delta^{18}$O in Fontainebleau was compared with that of the observations using monthly correlations against precipitation and vapour pressure deficit (VPD) from a nearby station (Daux et al., 2018). The monthly correlations and their bootstrapped significance were computed with the $dendroTools$ R package (Jevšenak, 2020).





The integrated growth-isotope responses simulated by ORCHIDEE and MAIDENiso models in Fontainebleau were qualitatively compared with observations using a bivariate response surface between tree-ring width variability and the joint variability of the two isotopes. A smoothed response surface was fitted using a data-adaptive bivariate generalized additive model (GAM)

implemented with the $mgcv$ package (Wood, 2017) in the R environment (R Core Team, 2020). This visualization device extends the dual isotope conceptual model of Scheidegger et al. (2000) to illustrate the mechanistic information content of the triple tree-ring constraint for models introduced in this study. It neatly reveals the complex association of tree growth with gas exchange inferred from stable isotopes in both observations and models.

To disentangle the relative importance of source water ($\delta^{18}O_{sw}$) and leaf water enrichment above source water ($\Delta^{18}O_{lw} = $

$\delta^{18}O_{lw} - \delta^{18}O_{sw}$) between May and August in determining the variability of $\delta^{18}O_R$ simulated by ORCHIDEE we used the Lindeman-Merenda-Gold (LMG) method (Grömping et al., 2006). It allows to quantify the contribution of different correlated regressors (here $\delta^{18}O_{sw}$ and $\Delta^{18}O_{lw}$) to the total $r^2$ of a multiple linear regression model.

## 3  Results

### 3.1  Model evaluation in Fontainebleau

#### 3.1.1  Tree-ring width and isotopic variability

ORCHIDEE shows a significant skill in simulating the interannual and multidecadal variability of oak tree-ring width ($r=0.59$, p<0.01) and latewood $\Delta^{13}C_R$ ($r=0.41$, p<0.01) and $\delta^{18}O_R$ ($r=0.49$, p<0.01) over the 20th century in Fontainebleau (Fig.1a–c). The magnitude of the interannual variability of $\delta^{18}O_R$ is well simulated (NSD=1.04) but that of tree-ring width is overestimated by 37% (NSD=1.37), while that of $\Delta^{13}C_R$ is underestimated by about a similar magnitude (NSD=0.55). $\Delta^{13}C_R$ is

systematically overestimated since 1980, when the observations show a decrease by about 1‰ (Fig. 1b). Overall, the model simulates 35% of observed tree-ring width variability and 17–24% of latewood isotopic variability over the 20th century.

The first-order autocorrelation or carrying over in observed tree-ring width is significant ($r_{lag1}=0.54$, p<0.001) and its magnitude indicates that ring width in the previous year explains up to 30% of current year tree-ring width variability. In contrast, simulated tree-ring width has no first-order autocorrelation ($r_{lag1}=0.02$, p>0.1) because ORCHIDEE does not account for the

carrying over in radial growth associated with carbon storage and remobilization. As a result, in years with extreme summer drought conditions like in 1921 and 1976 the model does not simulate any stem growth because photosynthesis is strongly suppressed (Fig. 1a). The lack of modulation of simulated radial growth by carbon storage dynamics makes the recovery after these extremes too fast compared with the observations. The first-order autocorrelation in the isotopic observations is significant only for $\Delta^{13}C_R$ ($r_{lag1}=0.37$, p<0.001) while in the simulations it is marginally significant for both $\Delta^{13}C_R$ ($r_{lag1}=0.18$,

p<0.1) and $\delta^{18}O_R$ ($r_{lag1}=0.27$, p<0.01).

The skill of ORCHIDEE compares well with that of the specialized MAIDENiso tree-ring model (Fig. 1a–c), which was specifically calibrated for the site over the period 1953–2000. Over this period, MAIDENiso is able to simulate between 30% and 46% of the total variability of the observations of tree-ring width and isotopes, which compares to the 20% to 44%





of the total observed variance simulated by ORCHIDEE over the same period with standard parameterization. MAIDENiso is considerably better than ORCHIDEE at simulating tree-ring width ($r$=0.68 vs $r$=0.51) and $\Delta^{13}C_R$ ($r$=0.58 vs $r$=0.45) variability, but despite simulating well the amplitude of $\Delta^{13}C_R$ and $\delta^{18}O_R$ (NSD=0.91–1.16) it substantially underestimates

the amplitude of tree-ring width (NSD=0.68). Unlike ORCHIDEE, it is able to simulate a significant first-order autocorrelation in tree-ring width ($r_{lag1}$=0.45, p<0.01) with a magnitude similar to that of the observations ($r_{lag1}$=0.50, p<0.01). This carrying over effect accounts for up to 25% of current year tree-ring width variability in the observations (1953–2000) and is thus an important component of the growth variability captured by its carbon remobilization dynamics.

MAIDENiso is able to simulate the observed decrease in $\Delta^{13}C_R$ since 1980 better than ORCHIDEE (Fig. 1b). The amount

and amplitude of $\delta^{18}O_R$ variability is slightly better simulated by ORCHIDEE ($r$=0.66, p<0.001; NSD=1.04) than by MAID-ENiso ($r$=0.54, p<0.001; NSD=0.91; Fig. 1c). The partitioning of the $r^2$ of a multiple linear regression shows that source water ($\delta^{18}O_{sw}$) and leaf water enrichment above source water ($\Delta^{18}O_{lw}$) account for 56% and 37% of the total variability of $\delta^{18}O_R$ simulated by ORCHIDEE over the period 1953–2000, respectively.

### 3.1.2   Observed and simulated relationships among tree-ring variables

So far the performance statistics used above only describe the unidimensional skill of the models and do not evaluate their ability to simulate the joint relationships that exist among tree-ring width and $\Delta^{13}C_R$ and $\delta^{18}O_R$. The strength of the correlations between observed tree-ring width and isotopic variability in Fontainebleau indicates that oak radial growth has 10% and 24% of common variance with latewood $\delta^{18}O_R$ and $\Delta^{13}C_R$, respectively (Fig. 1d). In turn, the isotopes have 16% of common variability. The interpretation of these three-way relationships in the observations can be aided by visualizing the bivariate surface

response of tree-ring width as a function of the dual latewood isotope variability (Fig. 1d). The resulting surface provides, at a glance, insights on causal and non-causal (indirect) relationships between environmental variability (temperature and drought stress) and stomatal responses and growth.

The bivariate surface response shows that in this site observed ring-width is positively and linearly related to latewood $\Delta^{13}C_R$ ($r$=0.48, p<0.01), whereas the relationship with $\delta^{18}O_R$ is negative ($r$=-0.31, p<0.05) and non-linear (Fig. 1d). This

indicates that narrow rings during dry years like in 1976 are mechanistically linked to reduced latewood $\Delta^{13}C_R$ because trees tend to close their stomata for longer during the summer to reduce water loss at the expense of reducing their photosynthesis. The opposite occurs during moist years, when growth, stomatal conductance and photosynthesis are high (Fig. 1d).

The apparent relationship between growth and $\delta^{18}O_R$ suggests interactions between the factors (air temperature, relative humidity and soil moisture) driving the growth process and $\delta^{18}O$ enrichment of source and leaf water (see axes in Fig. 1d). For

instance, years with high temperature tend to be also dry (e.g., 1976, 1990 and 1996) and associated with reduced atmospheric humidity and soil moisture, resulting in reduced growth and increased $\delta^{18}O$ enrichment of source water, vapour and leaf water due to higher evaporation rates and stomatal closure. The negative linear relationship between the two latewood isotopic variables ($r$=-0.40, p<0.01), apparent in the scatter of the observation points across the surface (Fig. 1d), provides evidence for a significant degree of stomatal control ($g_s$) of trees to avoid dehydration under warmer and drier conditions.





The two models simulate qualitatively different growth-isotope surface responses (Fig. 1e–f). ORCHIDEE simulates a surface response slightly more consistent with the observations than MAIDENiso, but none of the two simulations captures the observed non-linear relationship between tree-ring width and $\delta^{18}O_R$ that might arise from source water or isotopic enrichment

of leaf water. Although ORCHIDEE simulates a significant relationship of tree-ring width with $\delta^{18}O_R$ ($r=0.54$, p<0.001) and $\Delta^{13}C_R$ ($r=0.87$, p<0.001), the strength of the relationships is overestimated (Fig. 1e). Subtracting the isotopic variability of source water ($\delta^{18}O_{sw}$) from $\delta^{18}O_R$ ($\Delta^{18}O_R = \delta^{18}O_R - \delta^{18}O_{sw}$) to isolate the leaf signal does not change much the surface response of ORCHIDEE but increases substantially the correlation between the residual isotopic variability ($\Delta^{18}O_R$) and $\Delta^{13}C_R$ and GPP-driven ring-width (Fig. 2a–b).

MAIDENiso does not simulate any significant relationship between tree-ring width and $\delta^{18}O_R$ ($r=0.16$, p>0.1), but it captures well the magnitude of the observed correlation ($r=0.30$, p<0.01) between ring-width and $\Delta^{13}C_R$ (Fig. 1f). Both models largely overestimate the common variability between the isotopes, as is evident from the rather even spread of the data points across the diagonal of the surfaces (Fig. 1e–f). MAIDENiso and ORCHIDEE simulate 62% and 48% of common isotopic variability, respectively. This is three times above the observed 16%, indicating that the models overestimate the stomatal responses

to summer drought stress.

### 3.1.3   Climate response

The magnitude of the seasonal climatic responses of the observed tree-ring width, $\Delta^{13}C_R$ and $\delta^{18}O_R$ in Fontainebleau is well captured by the models, but there are some important differences in the timing and duration of the period of significant response (Fig. 3). The seasonal correlation patterns with precipitation and VPD indicate that tree-ring width and $\Delta^{13}C_R$ in

ORCHIDEE are too sensitive to moisture variability over the growing season compared with the observations (Fig. 3). In contrast, ring-width in MAIDENiso has too little sensitivity to precipitation variability. Nevertheless, MAIDENiso captures well the observed climatic response of $\Delta^{13}C_R$ to summer VPD, while the effect of summer VPD on $\delta^{18}O_R$ is better captured by ORCHIDEE and its isotopic forcing.

### 3.1.4   20th century change in water use efficiency

The $\Delta^{13}C_R$ data show that in Fontainebleau the observed intrinsic water use efficiency (iWUE) of oak measured as the change between 1901–1910 and 1990–2000 has increased by 25.5% following the gas-exchange scenario of constant $c_i/c_a$ (Fig. 4). Most of the change followed the steady increase in atmospheric $CO_2$ concentration since 1960, though the increasing trend in iWUE stalled around 1980 while atmospheric $CO_2$ continued increasing. ORCHIDEE produces a slightly lower increase in iWUE over the 20th century (21.2%), but also follows the gas exchange response of constant $c_i/c_a$. The model underestimates

iWUE since around 1980 and in contrast to the observations simulates a steady increase since 1960. This model-data mismatch is linked to the overestimation of $\Delta^{13}C_R$ in ORCHIDEE during this recent period as described earlier, an issue that does not affect MAIDENiso (Fig. 1b).





### 3.1.5 Simulated relationship between productivity and isotopic variability

Simulated isotopic variability in Fontainebleau is significantly correlated with growing season (May-August) GPP (Fig. 4b). The magnitude of the correlations indicate that simulated $\delta^{18}O_R$ ($r$=-0.62, p<0.001) and $\Delta^{13}C_R$ ($r$=0.84, p<0.001) explain 38% and 71% of GPP variability over the 20th century, respectively. Since stem growth in ORCHIDEE depends directly on the allocation of GPP, simulated tree-ring width variability is by definition strongly correlated with growing season GPP ($r$=0.90, p<0.001) as it can be seen in Fig. 4b.

### 3.2 Model performance across sites

The performance of ORCHIDEE for tree-ring width, $\Delta^{13}C_R$ and $\delta^{18}O_R$ varies substantially across sites (Fig. 5), with no clear pattern along the climate gradient from Finland to France or between species for any parameter. However, it is clear that the isotopic variability is better simulated than tree-ring width and also that $\delta^{18}O_R$ is the tree-ring variable best simulated by the model. Tree-ring width is well simulated (25–30% of the observed variability) at only two out of six sites. It is not clear whether the fact that the best simulations are for the two southernmost sites (Fontainebleau and Annnecy) is a coincidence or suggest that the model processes and/or parameters are biased in favour of deciduous temperate forests. In the remaining sites the simulations account for less than 10% ($r$<0.32) of the observed variability. The lower ability of ORCHIDEE to simulate tree-ring width is to a large extent due to the present inability of the model to simulate the significant carrying over effect of growth (autocorrelation) evident in the observations (Fig. 7a).

Although ORCHIDEE is able to simulate about 10–64% ($r$=0.31–0.80) of the observed $\Delta^{13}C_R$ variability, it tends to underestimate its amplitude by 30–60%, particularly in the northernmost sites of pine (Kessi and Sivakkovaara) where the isotopic ratios were measured over the whole ring (Fig. 5). With the exception of the northernmost site, the amplitude of $\delta^{18}O_R$ variability is simulated within ±20% and the simulations account for 13–55% ($r$=0.36–0.74) of the variance of the observations. The primary driver of simulated $\delta^{18}O_R$ variability in five out of the six sites is the isotopic composition of source water ($\delta^{18}O_{sw}$). It accounts for 46–85% of the simulated $\delta^{18}O_R$ variability, whereas leaf water enrichment ($\Delta^{18}O_{lw}$) accounts for 14–53% of $\delta^{18}O_R$ (Fig. 6a). Except by the coolest northernmost site, where $\delta^{18}O_{sw}$ drives almost entirely $\delta^{18}O_R$ (85% of the variance), there is no clear latitudinal pattern for the relative importance of $\delta^{18}O_{sw}$ or $\Delta^{18}O_{lw}$.

ORCHIDEE overestimates the correlation between $\delta^{18}O_R$ and $\Delta^{13}C_R$ in the temperate sites in France, but it simulates very well the magnitude of the isotopic coupling observed in the boreal oak and pine sites in Finland (Fig. 6b). This means that the simulated stomatal control and responses to atmospheric humidity are overestimated in the temperate deciduous PFT, as is apparent in a stronger correlation between simulated isotopic variability and VPD in these sites compared with the observations (Fig. 6c–d).

The $\delta^{18}O_R$ simulations of the LPX-Bern global model are systematically better than the simulations of ORCHIDEE in terms of correlations and amplitude of the interannual variability (Fig. 5). In Fontainebleau LPX-Bern simulates 74% ($r$=0.86) of the observed $\delta^{18}O_R$ variability. In this site, ORCHIDEE and MAIDENiso are able to simulate only 40% ($r$=0.63) and 34% ($r$=0.58) of the observed variability, respectively. Three sites (Kessi, Sivakkovaara and Fontainebleau) exceed the range of





performance of ORCHIDEE (blue shading in Fig. 5), mostly due to the the good skill of the LPX-Bern simulation in terms of correlation with the observations.

The significant relationship found between simulated $\delta^{18}O_R$ and GPP in Fontainebleau is also observed across all the other
sites (Fig. 8a). Its strength ($r$=-0.47 to -0.73) appears to increase from the boreal to the temperate region, accounting for up to 22–53% of GPP variability. This empirical relationship is driven by a synergistic effect of source water and leaf water enrichment above source water because both variables correlate negatively with GPP (Fig. 8b). Leaf water enrichment tends to correlate with GPP more strongly than source water, though in the northernmost pine site only source water is correlated with GPP. $\Delta^{13}C_R$ is also significantly correlated with GPP in most of the sites, but correlations are insignificant or change sign in
the colder northernmost sites (Fig. 8a). Tree-ring width is by definition highly correlated with GPP (Fig. 8a).

## 4   Discussion

### 4.1   Integrating tree-ring width and carbon and oxygen isotopes for insights on growth and gas exchange

The enhanced information content obtained by combining multiple and complementary tree-ring variables has long been recognized in multi-proxy dendroclimatology (McCarroll et al., 2003; Loader et al., 2008; Hilasvuori et al., 2009; Daux et al.,
2011; Schollaen et al., 2013; Loader et al., 2015) and dendroecology (Guerrieri et al., 2009; Savard, 2010; Leonelli et al., 2012; Shestakova and Martínez-Sancho, 2020), but its potential remained largely untapped in ecological modelling. Tree-ring widths provide a historical record of annual aboveground biomass increment (Clark et al., 2001; Bouriaud et al., 2005; Babst et al., 2018; Cernusak and English, 2015; Foster et al., 2016; Dye et al., 2016; Evans et al., 2017; Shestakova et al., 2019). The stable carbon isotope ratio of plant material, usually reported as $\delta^{13}C$, is related to the ratio of intercellular ($c_i$) and atmospheric $CO_2$
concentration (Farquhar et al., 1982). As this ratio varies according to changes in stomatal aperture ($g_s$; supply of $CO_2$) and assimilation rate ($A$; $CO_2$ demand), it integrates the physiological response of plants to environmental changes such as drought stress and increasing atmospheric $CO_2$ concentration (McCarroll and Loader, 2004; Gessler et al., 2014).

Changes in $c_i$ can result from changes in either $A$ or $g_s$, thus the interpretation of changes in carbon isotope alone is difficult. The imprint of leaf evaporative enrichment on $\delta^{18}O$ of tree-ring cellulose is not affected by $A$ and like the carbon
isotope depends on $g_s$ (McCarroll and Loader, 2004; Gessler et al., 2014). Hence, the leaf enrichment signal of tree-ring $\delta^{18}O$ ($\Delta^{18}O_{lw}$) is an independent proxy for $g_s$ and can be used to disentangle the physiological drivers ($A$ or $g_s$) of variations in carbon isotope fractionation and growth in the same tree ring (Saurer et al., 1997; Scheidegger et al., 2000; Barnard et al., 2012).

The concurrent variability of tree-ring width, $\Delta^{13}C_R$ and $\delta^{18}O_R$ is driven by coordinated ecophysiological responses to
external environmental factors that, depending on species and location, can imprint a varying degree of covariability among them as seen in the tree-ring triplet (Fig. 1d). In a temperate location such as Fontainebleau, it is expected that dry summer conditions associate with narrow tree rings, decreased $\Delta^{13}C_R$ and increased oxygen isotope enrichment in the leaf water (thus higher $\delta^{18}O_R$) because trees close their stomata to reduce $g_s$ and avoid dehydration during drought stress at the expense of photosynthesis (Saurer et al., 1997; Scheidegger et al., 2000; Barnard et al., 2012). Consistent with this expectation, our





bivariate response surface shows that years with narrow rings like 1976, 1990 and 1996 are associated with low $\Delta^{13}C_R$ and high $\delta^{18}O_R$ in latewood (Fig. 1d). The opposite occurs during moist years with wider rings, when conditions for stomatal aperture and photosynthesis are optimal. This demonstrates that projecting tree-ring width variability into the isotopic space allows unravelling the underlying mechanistic relationships between tree growth and gas exchange to obtain an integrated picture of how trees respond to drought stress, an eventually also to temperature anomalies if $\delta^{18}O_R$ can be interpreted as a temperature proxy (Fig. 1d–f).

The often dominant effect of $\delta^{18}O_{sw}$ on $\delta^{18}O_R$ variability (Fig. 6a) dilutes the leaf enrichment signal and reduces the correlation between $\delta^{18}O_R$ and $\Delta^{13}C_R$ (Fig. 2a). Studies using the dual isotope conceptual model to interpret physiological signals typically remove or minimize its effect (Scheidegger et al., 2000; Barnard et al., 2012; Roden and Siegwolf, 2012). In modelling studies, this signal can be quantified and used to evaluate the quality of the isotopic forcings (precipitation and vapour) of modelled $\delta^{18}O_R$ and attribute data-model mismatches. In addition, $\delta^{18}O_{sw}$ carries a well-known paleo-temperature signal (McCarroll and Loader, 2004) that can be used to disentangle the climatic forcing of past growth anomalies and physiological responses inferred from $\Delta^{13}C_R$ well beyond the duration of the meteorological records.

The three-way correlations among tree-ring width and $\Delta^{13}C_R$ and $\delta^{18}O_R$ (or $\Delta^{18}O_R$) variability provide a measure of the interannual coupling between growth and gas exchange, revealing direct and indirect associations driven by common or correlated environmental forcings (Fig. 1d). The strength of the direct causal relationships expected from mechanistic understanding, like the correlation between isotopes (Scheidegger et al., 2000) and between photosynthesis or biomass increment and carbon isotopes (Belmecheri et al., 2014; Battipaglia et al., 2013; Lévesque et al., 2014; Fernandes et al., 2016), is a simple benchmark to constrain the physiological parameterization of models as we discuss below.

## 4.2 Critical processes for the concurrent simulation of multiple tree-ring variables in global land surface models

Our evaluation in Fontainebleau and other five sites across a boreal-to-temperate climate gradient in Europe showed that ORCHIDEE simulates better the interannual variability of tree-ring stable isotopes than ring-width (30-64% and <30% of the observed variability, respectively), with a general performance for the stable isotopes similar to MAIDENiso and LPX-Bern models (Figs.1 and 5). The lower skill for tree-ring width results from the inability of ORCHIDEE to simulate the significant temporal autocorrelation observed in ring-width variability (Fig. 7a). This temporal carrying over effect is common in tree-ring width (Cailleret et al., 2018; Breitenmoser et al., 2014) and results from carbon remobilization from previous years (Kagawa et al., 2006) and to some extent from cambial dynamics (Vaganov et al., 2011). It varies considerably with species and location but typically accounts for 20–25% of current year ring-width variability (Breitenmoser et al., 2014).

The ORCHIDEE version used in this study (r898) lacks the representation of the influence of carbon storage on simulated tree-ring width and instead growth reflects only current year GPP variability (see Figs. 3b and 6a). This simplified representation of growth is currently the major limitation of the model to simulate the tree-ring triplet (Fig. 5). Nevertheless, in Fontainebleau ORCHIDEE still simulates 26% of the observed interannual variability but overestimates its amplitude and the speed of recovery from drought extremes like 1976 (Fig. 1a). Legacy effects of reduced growth following drought events can persist for 1 to 4 years in drought-sensitive ecosystems (Anderegg et al., 2015; Cailleret et al., 2018). Such prolonged legacy





effects are typically not simulated by global carbon cycle models because most of them, like ORCHIDEE, lack the representation of the significant dependency of interannual tree growth on carbon remobilization from storage in their carbon allocation schemes (Anderegg et al., 2015).

Unlike global models, MAIDENiso explicitly represents the autocorrelation in tree-ring width in its carbon allocation scheme (Misson, 2004; Danis et al., 2012) and as a result it was able to capture the enduring effect of the extreme drought of 1976 in Fontainebleau even when tree growth is represented through GPP allocation as in ORCHIDEE (Fig. 1a). This result demonstrates that a simple approach to represent the dependency of tree growth on carbon remobilization might produce a substantial improvement in ORCHIDEE and allow a direct tree-ring constraint for the simulation of ecosystem recovery from climate ex-

tremes. Implementing wood formation dynamics would be a fully process-based approach to represent tree-ring growth (Fritts et al., 1999; Friend et al., 2019), but in a global model it implies an important tradeoff with computing time.

    ORCHIDEE was able to simulate 30-64% of the observed $\Delta^{13}C_R$ variability along the climate gradient but the amplitude of the interannual variations was underestimated by 30-60% (Fig. 5) and the observed decrease in $\Delta^{13}C_R$ since 1980 in Fontainebleau was not captured (Fig. 1b). An earlier study with ORCHIDEE also found a similar underestimation of the

interannual variability of $\Delta^{13}C_R$ for larch in northeastern Yakutia, where the model simulated 26% ($r = 0.51$) of the observed variability (Churakova et al., 2016). The low amplitude of the simulated variability can be related to missing fractionation and mixing processes and the parameterization of soil and processes that affect $c_i$ such as photosynthesis and moisture responses.

    A simple parameter sensitivity test in Fontainebleau (not shown) indicated that the amplitude of simulated $\Delta^{13}C_R$ is very sensitive to soil depth and photosynthetic capacity (Vcmax). This suggests that simulated $\Delta^{13}C_R$ and the associated drought

and stomatal responses can be better parameterized using tree-ring data. Nevertheless, besides an improved paramaterization, using a more complete formulation for carbon discrimination combined with a scheme of carbohydrate mixing (Hemming et al., 2001; Ogée et al., 2009; Danis et al., 2012) should contribute to improve the simulation of $\Delta^{13}C_R$ and stomatal responses in ORCHIDEE.

    The long-term evaluation in Fontainebleau shows that ORCHIDEE simulates no overall change in 20th century $\Delta^{13}C_R$ (Fig

1b), implying that the simulated $c_i/c_a$ ratio remained roughly constant as atmospheric $CO_2$ concentrations increased by about 25% (Saurer et al., 2004). Under this type of gas-exchange response, Eq. (9) shows that iWUE should increase proportionally to the relative increase in atmospheric $CO_2$ (Saurer et al., 2004). Indeed, the simulated centennial increase in iWUE is 21.2% with respect to the 1901-1910 period (Fig. 4a). This is still comparable with the observed increase in iWUE of 25.5%, indicating that the decrease of about 1‰ in $\Delta^{13}C_R$ (decrease in $c_i$) since 1980 (Fig 1b) was not sufficient to shift the constant $c_i/c_a$ type

of response of oak to rising $CO_2$ in Fontainebleau (Fig. 4a). Constant $c_i/c_a$ ratio over the 20th century is the most common physiological response reported for trees in Europe (Saurer et al., 2014; Frank et al., 2015) and has also been correctly simulated by the LPX-Bern vegetation model (Saurer et al., 2014; Keller et al., 2017), which uses the same formulation for $\Delta^{13}C_R$ (Eq. 2) than ORCHIDEE. The implication of this stomatal response for land-atmosphere interactions is a centennial-scale reduction in transpiration as trees under present environmental conditions use less water for the production of the same amount of biomass compared with earlier decades of the 20th century.





ORCHIDEE simulated the amplitude of $\delta^{18}O_R$ within $\pm 20\%$ and accounted for 13-55% of the observed variability (Fig. 5). Consistent with earlier palaeoclimatic and modelling studies (Roden et al., 2000; Robertson et al., 2001; Treydte et al., 2014; Danis et al., 2006, 2012; Keel et al., 2016), most of the simulated variability of $\delta^{18}O_R$ (46–85%) in ORCHIDEE is driven by

5 $\delta^{18}O_{sw}$ (Fig. 6a). Leaf water enrichment ($\Delta^{18}O_{lw}$) accounted for 14–53% of $\delta^{18}O_R$ variability, without showing any latitudinal pattern in its contribution (Fig. 6a). Nevertheless, the coolest northernmost pine site had the lowest contribution of $\Delta^{18}O_{lw}$ to $\delta^{18}O_R$ variability as it would be expected from lower transpiration rates in a cool environment (Treydte et al., 2014).

The general pattern of relative contributions of source and leaf water enrichment highlights the strong dependence and sensitivity of $\delta^{18}O_R$ to the choice of the precipitation isotopic forcing in modelling studies. Differences in the isotopic drivers

10 might account for most of the observed differences among models (Fig. 5). The isotopic forcings for LPX-Bern (Haese et al., 2013; Keel et al., 2016) and ORCHIDEE (Risi et al., 2010) were produced by isotope-enabled global circulation models driven or nudged by reanalysis products, whereas for MAIDENiso simple regressions with temperature and precipitation were used to produce daily precipitation and water vapour $\delta^{18}O$ forcings (Danis et al., 2012).

The LPX-Bern simulations of $\delta^{18}O_R$ consistently compared better with the observations than the simulations of ORCHIDEE

15 and MAIDENiso, both in terms of correlations and amplitude of the variability (Fig. 5). Notably, LPX-Bern was able to simulate 74% of the observed $\delta^{18}O_R$ variability in the grid-box corresponding to Fontainebleau, which is considerably higher than the variance accounted for by ORCHIDEE (44%) and MAIDENiso (29%) in the site. This pattern suggests that the higher performance of LPX-Bern might be related to a better isotopic forcing.

Overall, a scheme of carbon storage and remobilization dynamics (e.g., Misson, 2004) or wood formation (Fritts et al., 1999;

20 Friend et al., 2019) should be implemented in ORCHIDEE as part of the ongoing developments to produce novel observational benchmarks from tree-ring width data (Jeong et al., 2020). Such development will allow capturing the significant autocorrelation in tree-ring width (Fig. 7a) and improving the simulation of $\Delta^{13}C_R$ variability, resulting in a better representation of the impacts of climate extremes on forest ecosystems. Since the choice of isotopic forcing has a large impact on the simulation of $\delta^{18}O_R$, more than one forcing should be used to evaluate the magnitude of the uncertainty of source water signals in simulated

25 $\delta^{18}O_R$.

### 4.3 Constraining model processes with the growth-isotope tree-ring triplet

The novel simulation of the growth-isotope tree-ring triplet (ring width, $\Delta^{13}C_R$ and $\delta^{18}O_R$) in a global land surface model enabled us to use known mechanistic relationships between isotopes (Scheidegger et al., 2000) and between growth and carbon isotopes (Francey and Farquhar, 1982; Cernusak and English, 2015; Shestakova et al., 2019) to benchmark the physiological

30 responses of the model beyond the traditional use of the interannual variability or trends (Panek and Waring, 1997; Danis et al., 2012; Bodin et al., 2013; Churakova et al., 2016; Keel et al., 2016; Keller et al., 2017; Ulrich et al., 2019). A much stronger negative correlation between simulated $\Delta^{13}C_R$ and $\delta^{18}O_R$ than in the observations suggests that ORCHIDEE might overestimate the limitation of photosynthetic carbon assimilation by stomatal control in the temperate region (Rennes, Fontainebleau and Annecy; Fig. 6b).



Removing the $\delta^{18}O_{sw}$ effect from simulated $\delta^{18}O_R$ to highlight the effect of leaf water enrichment in Fontainebleau further increases the strength of the isotopic coupling (Fig. 2). It implies that growth and gas exchange for the temperate Broadleaf Summergreen PFT of the model are likely too sensitive to atmospheric evaporative demand and drought stress. This is consis-

tent with correlations between VPD and simulated isotopic variability being stronger than in the observations in these sites (Fig. 6c–d), and with an overestimated sensitivity of simulated tree-ring width to precipitation and VPD in Fontainebleau (Fig. 3). In the boreal region (Kessi, Sivakkovvara and Bromarv), ORCHIDEE simulated a level of correlation between isotopes (Fig. 6b) and sensitivity to VPD (Fig. 6c) in good agreement with the observations, indicating a better gas-exchange parameterization for the boreal Broadleaf Summergreen and Needleleaf Evergreen PFTs. These results illustrate how tree-ring data can be used

to evaluate and identify critical processes in the parameterization of a global land surface model. The parameterization of the temperate PFT can be improved by assimilating tree-ring data together with other ecosystem observations at shorter time scales using a data assimilation technique (Peylin et al., 2016; Thum et al., 2017).

We did not remove the effect of $\delta^{18}O_{sw}$ on $\delta^{18}O_R$ in the observations because the actual isotopic composition of precipitation is unknown. Thus, we caveat that the interpretation of the direct correlations between the two isotopes (Fig. 1) might violate

the basic assumption of a dominant leaf-water signal in the dual-isotope approach (Roden and Siegwolf, 2012). Nevertheless, removing simulated $\delta^{18}O_{sw}$ from $\delta^{18}O_R$ reduces the correlation between the isotopes in the observations for the temperate sites and further increases the correlation in the simulations (not shown), reinforcing the finding that ORCHIDEE overestimates stomatal control in the temperate PFT. The strengthening of the isotopic coupling after removing the effect of $\delta^{18}O_{sw}$ from simulated $\delta^{18}O_R$ can be seen in Fontainebleau (Fig. 2).

Observed tree-ring width was significantly and positively correlated with $\Delta^{13}C_R$ in Fontainebleau (Fig. 1d). This correlation is common in pine and oak forests in the mid-latitudes of Europe and reflects that tree growth and leaf physiology are both similarly driven by water stress (Shestakova et al., 2019). ORCHIDEE (r898) largely overestimated this relationship because of an excessive drought sensitivity and lack of the representation of carbon remobilization, which would buffer the effect of the instantaneous leaf response on tree growth (Fig. 1e). Because of the latter process MAIDENiso captured the right magnitude

of the expected coupling between leaf physiology and radial growth in the site (Fig. 1f).

## 4.4 Simulated relationship between isotopic variability and photosynthesis

Our results showed that the simulated interannual variability in $\delta^{18}O_R$ was significantly and negatively correlated with GPP across all sites in the climate gradient ($r$ = -0.47 to -0.73, p < 0.01) because both source water ($\delta^{18}O_{sw}$) and leaf water enrichment ($\Delta^{18}O_{lw}$) related negatively with GPP (Fig. 8). Although $\Delta^{18}O_{lw}$ dominates the correlation in most sites, it is the

synergistic effect of $\delta^{18}O_{sw}$ variations that contributes to the apparent spatial consistency of the $\delta^{18}O_R$–GPP correlation. The negative sign of the correlations of GPP with both components of $\delta^{18}O_R$ points to temperature as their underlying cause. On the one hand, higher temperatures during the growing season result in enriched $\delta^{18}O$ in precipitation and further enrich $\delta^{18}O_{sw}$ through evaporation. On the other hand, higher temperatures also lead to increased VPD, which increases leaf water enrichment and reduces GPP because of stomatal closure (Fig. 1d).





An earlier study attributed the correlation between $\delta^{18}O_R$ variability and satellite-based estimates of Net Primary Productivity (NPP) at large spatial scales to the common but opposite effect of VPD on leaf water enrichment and NPP, without considering the possible effect of source water (Levesque et al., 2019). Our finding suggests that $\delta^{18}O_R$-productivity correlations should be interpreted with caution because they are not entirely driven by leaf physiology, and source water is often a dominant driver of $\delta^{18}O_R$ (Fig. 6a).

The simulated interannual variability in $\Delta^{13}C_R$ also correlated with GPP in most sites, but the relationship was less spatially consistent since its significance and sign varied across the boreal-to-temperate climate gradient (Fig. 8a). A positive correlation between simulated $\Delta^{13}C_R$ and GPP in all temperate and boreal oak sites ($r = 0.45$ to $0.85$, $p < 0.01$) is coherent with a stomatal limitation of carbon assimilation (Shestakova et al., 2019). In the two boreal pine sites, correlations were weaker and only the northernmost site had a significant negative correlation ($r = -0.40$, $p < 0.01$). In such cool and moist environment, internal leaf $CO_2$ concentration is controlled by photosynthetic rate, which is limited by temperature and sunshine (McCarroll and Loader, 2004; Hilasvuori et al., 2009).

This contrast in $\Delta^{13}C_R$–GPP correlations between the temperate and boreal region mirrors a shift from a positive to a negative relationship between $\Delta^{13}C_R$ and tree-ring width due to a change from water-limited to temperature-limited leaf physiology (Shestakova et al., 2019). These model-based isotope-GPP relationships are consistent with recent findings showing that tree-ring $\delta^{18}O$ correlates with satellite-based NPP over large spatial scales, whereas $\Delta^{13}C$ has a more local and variable correlation with GPP (Belmecheri et al., 2014; Levesque et al., 2019; Tei et al., 2019).

## 5 Conclusion

We demonstrated the potential of tree-ring width and carbon and oxygen stable isotopes to constrain the representation of tree growth and physiology in the global land surface model ORCHIDEE (r898), bridging the long-standing gap between the tree-ring and land surface modelling communities. ORCHIDEE had an overall performance to simulate tree-ring isotopic variability similar to that of the specialized MAIDENiso tree-ring model and the global vegetation model LPX-Bern. However, the lack of representation for the dependency of tree growth and carbon isotopes on carbon remobilization from storage in the model is currently a major limitation for the concurrent simulation of the ring width-isotope triplet (ring width, $\Delta^{13}C_R$ and $\delta^{18}O_R$). The large contribution of the isotopic signature of source water on $\delta^{18}O_R$ makes its simulation sensitive to the choice of the prescribed isotopic drivers. Future modelling studies would benefit from quantifying uncertainties on $\delta^{18}O_R$ variability from the isotopic drivers.

The simulated long-term physiological response of constant leaf $c_i/c_a$ ratio under rising $CO_2$ during the 20th century is consistent with the observations in the temperate region, despite an overestimation of seasonal drought stress and the limitation of photosynthetic carbon assimilation by stomatal control. The interannual variability of productivity (GPP) correlated consistently with the simulated variability of tree-ring carbon and oxygen stable isotopes, but correlations with oxygen isotopes should be interpreted with caution because of the significant and often large effect of source water on the correlations. These results establish the foundations for improving the simulation of tree rings in ORCHIDEE (Jeong et al., 2020) and a forthcom-





ing optimization of model parameters using tree-ring data together with other short-term ecosystem observations (e.g., remote sensing, eddy-covariance, forest inventories) in a formal data assimilation technique (Peylin et al., 2016). Such advances should contribute to reduce current uncertainties in historical and future trends in forest carbon and water cycling.

5 *Data availability.* All the tree-ring data and ORCHIDEE tree-ring simulations used in this study are available from the corresponding author upon request.

*Code and data availability.* The model code (SVN r898) is accessible in the ORCHIDEE webpage https://forge.ipsl.jussieu.fr/orchidee/wiki/SourceCode. The scripts required for reproducing the intermediate results and figures are available at https://github.com/campsidium/orchidee_biogeo

*Author contributions.* JB, PP, TL and VD designed the research. TL conducted the simulations with support of CR. JJ and SL contributed
10 to model development and research design. JB revised the model, analyzed the data and wrote the manuscript. All the authors contributed to data interpretation and commented on the manuscript.

*Competing interests.* The authors declare no competing interests.

*Acknowledgements.* This research is part of a post-doctoral grant to Jonathan Barichivich from the presidential programme "Make Our Planet Great Again" (MOPGA) conducted through the Centre National de la Recherche Scientifique (CNRS) of France. We thank Joel Guiot
15 for sharing the simulations of MAIDENiso for Fontainebleau and Manuel Gloor and Roel Brienen for discussion. We acknowledge support from the European Commission, Horizon 2020 Framework Programme, VERIFY (grant no. 776810).





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





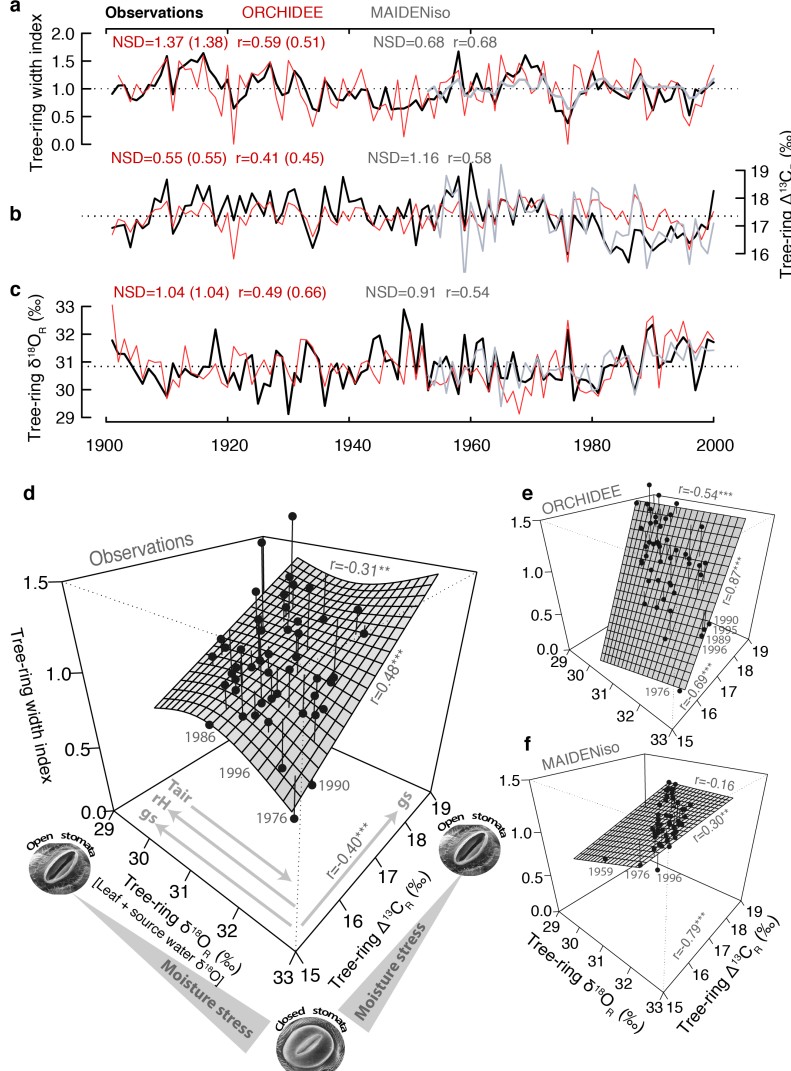

**Figure 1.** Comparison of simulated and observed tree-ring width and isotopic variability in Fontainebleau and the triple tree-ring constraint to assess the integrated model responses. **a-c** Comparison of tree-ring width and latewood $\Delta^{13}C_R$ and $\delta^{18}O_R$ variability simulated by ORCHIDEE (red) and MAIDENiso (gray) with observations (black). The normalised standard deviation (NSD) and Pearson correlation (r) are indicated in each case. The mean of the simulations was set to that of the observations for each parameter. **d-f** Bivariate response surface of tree-ring width as a function of $\delta^{18}O_R$ and $\Delta^{13}C_R$ variability based on observations and on ORCHIDEE and MAIDENiso simulations over the common period 1953–2000. The individual years are indicated by the black dots and their vertical distance to the surface is represented by the lines. The most extreme years with low growth and high moisture stress are labelled in each panel. The three-way correlations between the tree-ring variables are indicated along the edges of each surface: tree-ring width vs $\delta^{18}O_R$ (top), tree-ring width vs $\Delta^{13}C_R$ (right) and $\delta^{18}O_R$ vs $\Delta^{13}C_R$ (bottom). The asterisks denote the significance levels of the correlations: *: p< 0.1, **: p< 0.05 and ***:p< 0.001. In panel **d**, the relationship between stomatal aperture and moisture stress is indicated along the isotopic axes together with the expected changes in stomatal conductance ($g_s$) and relative humidity ($rH$) according to the dual isotope model (Scheidegger et al., 2000). The expected enrichment of source water $\delta^{18}O$ with increasing air temperature ($Tair$) is also indicated.





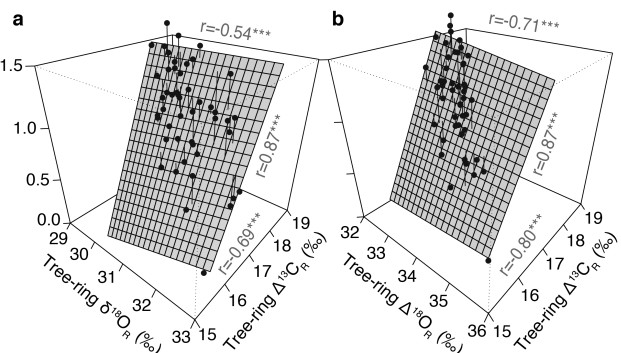

**Figure 2.** Comparison of bivariate response surfaces simulated by ORCHIDEE with and without the influence of source water ($\delta^{18}O_{sw}$) over the common period 1953–2000. **a** Response surface with $\delta^{18}O_R$ that combines the effect of source water and leaf water enrichment as in Fig. 1e. **b** Response surface with $\Delta^{18}O_R$ ($\delta^{18}O_R - \delta^{18}O_{sw}$), which removes the effect of source water from $\delta^{18}O_R$ but conserves the effect of leaf water enrichment and associated stomatal responses to air relative humidity (Scheidegger et al., 2000). The significance of the three-way correlations between the tree-ring variables are indicated along the edges of each surface as in Fig. 1.

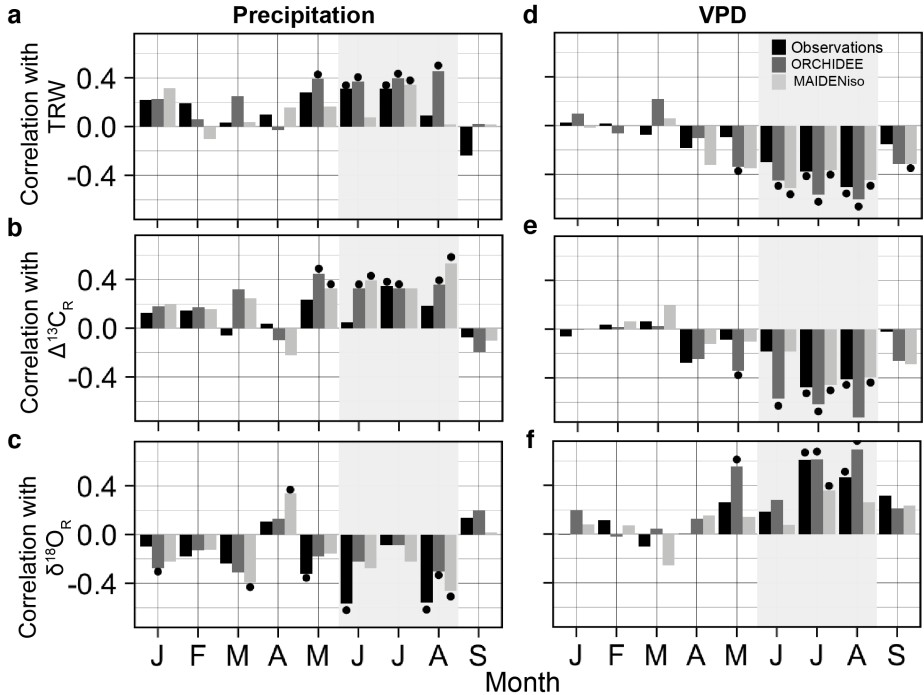

**Figure 3.** Monthly climatic response functions of the observations in Fontainebleau and the corresponding tree-ring parameters simulated by ORCHIDEE and MAIDENiso over the period 1960–2000. **a–c** Correlation between monthly precipitation from January to September and tree-ring width, $\Delta^{13}C_R$ and $\delta^{18}O_R$. **e–f** Same than in **a–c** but for VPD (air vapour pressure deficit). Significant correlations at the 95% confidence level are indicated by a black dot. The gray shading denotes summer months.

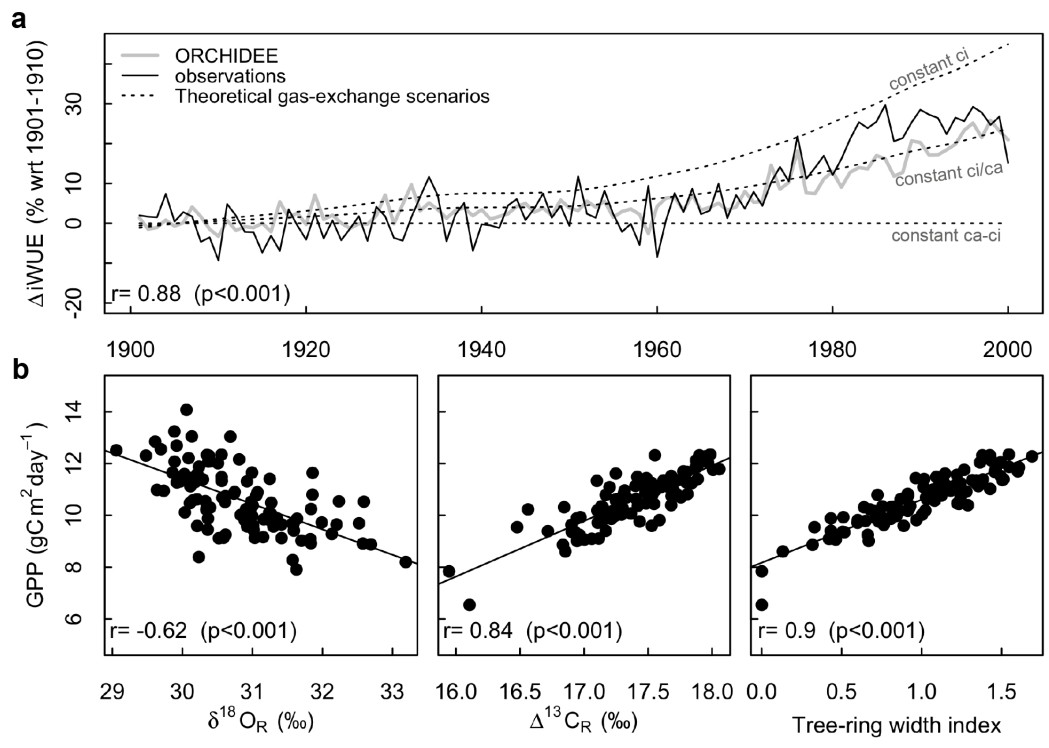

**Figure 4.** Comparison of observed and simulated changes in iWUE and relationships between simulated tree-ring variables and productivity during the 20th century in Fontainebleau. **a** Observed and modelled change in iWUE with respect to the mean of the earlier period 1901–1910. The theoretical scenarios of gas exchange of Saurer et al. (2004) are shown as reference, considering mean $c_i$ over 1901–1910 as starting point. **b** Correlations between simulated May-August Gross Primary Productivity (GPP) and simulated $\delta^{18}O_R$ (left), $\Delta^{13}C_R$ (middle) and tree-ring width (right) over the period 1901–2000.



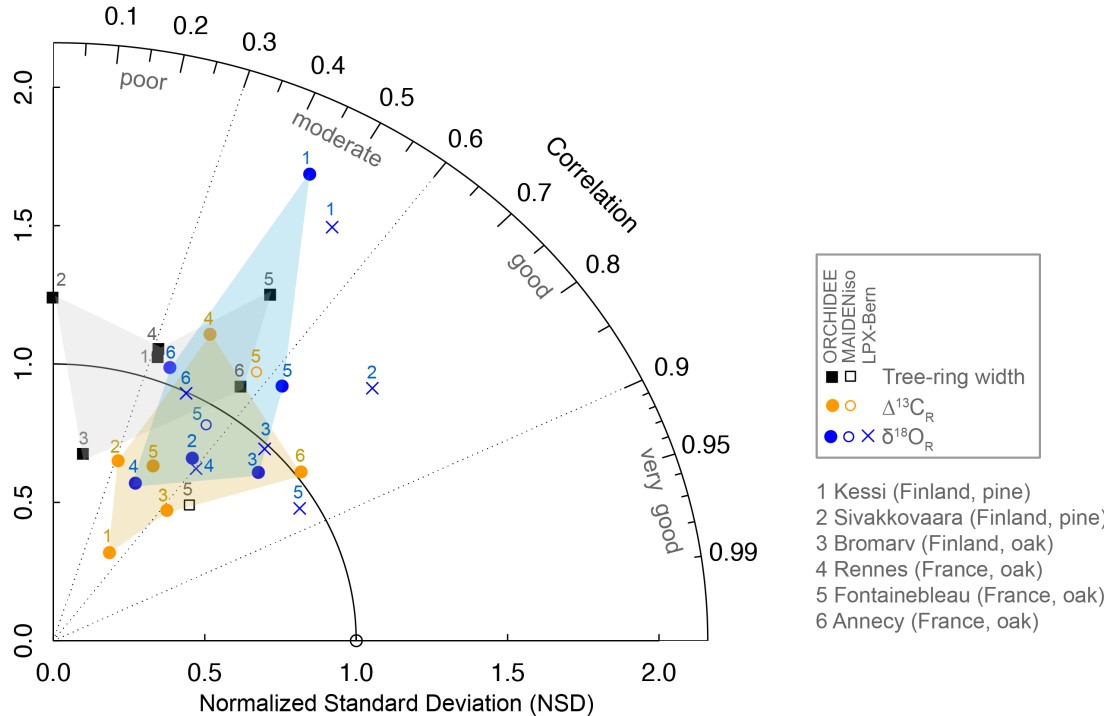

**Figure 5.** Taylor diagram showing the correlation (angular coordinate) and normalized standard deviation (NSD; radial coordinate) of the simulated tree-ring parameters (ORCHIDEE, MAIDENiso and LPX-Bern) with respect to the observations for the six sites used in this study (see Table 1). Tree-ring width is denoted by square symbols and $\Delta^{13}C_R$ and $\delta^{18}O_R$ by orange and blue circles, respectively. The models are denoted by solid symbols (ORCHIDEE), open symbols (MAIDENiso in Fontainebleau or site 5) and blue crosses (LPX-Bern). The statistics were computed over the common period 1960–2000, except in Annecy (site 6) where data availability limited the comparison to the period 1971–2000. The target point (NSD and correlation equal to 1) is represented by a circle. The gray, blue and orange shading denotes the range of the performance statistics covered by ORCHIDEE for tree-ring width, $\Delta^{13}C_R$ and $\delta^{18}O_R$, respectively.

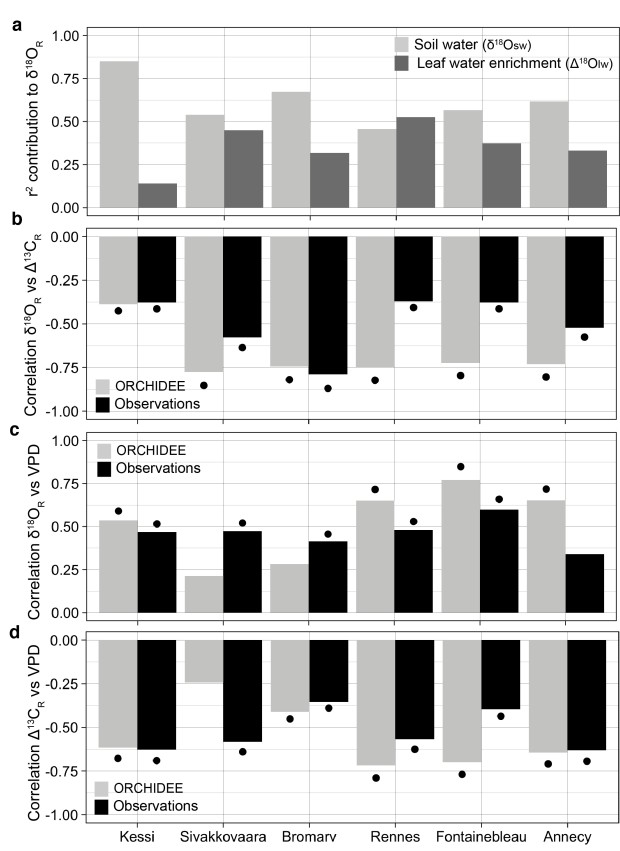

**Figure 6.** Influence of source and leaf water $\delta^{18}O$ on simulated $\delta^{18}O_R$ in ORCHIDEE and response of isotopic variability to VPD across the climate gradient from boreal Finland (left) to temperate France (right) during 1960–2000. **a** Relative contribution ($r^2$) of soil (light gray) and leaf water (dark gray) $\delta^{18}O$ to simulated $\delta^{18}O_R$ variability. **b** Correlation between $\delta^{18}O_R$ and $\Delta^{13}C_R$ in simulations (dark gray) and observations (black). **c–d** Correlations between growing season VPD and simulated (dark gray) and observed (black) $\delta^{18}O_R$ and $\Delta^{13}C_R$ variability, respectively. For Annecy the period of analysis is 1971–2000. Significance at the 95% confidence level is indicated by the black dots.





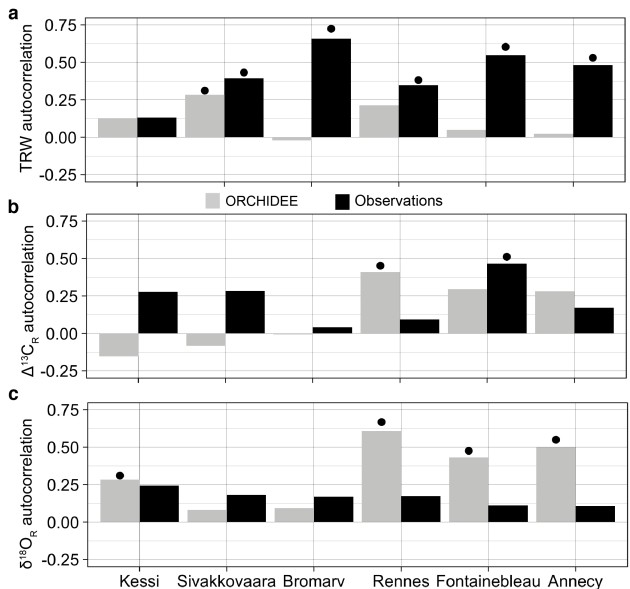

**Figure 7.** First-order autocorrelation in observed (black) and simulated (gray) tree-ring variables over the period 1960–2000. **a** Tree-ring width. **b** $\Delta^{13}C_R$. **c** $\delta^{18}O_R$. For Annecy the period of analysis is 1971–2000. Significance at the 95% confidence level is indicated by the black dots.

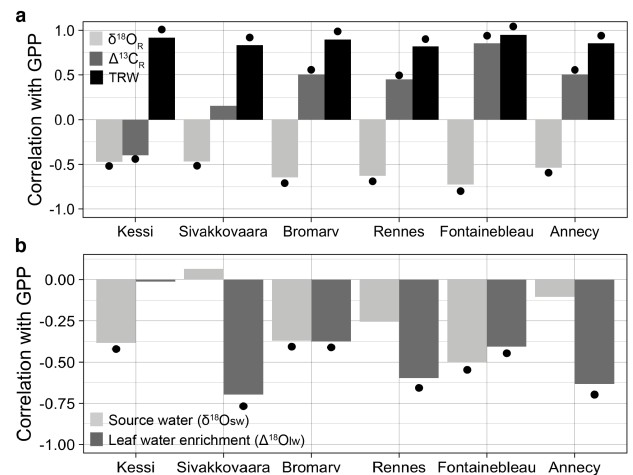

**Figure 8.** Correlations between the simulated tree-ring variables at each site and simulated May-August Gross Primary Productivity (GPP) over the evaluation period 1960–2000. **a** Correlations of $\delta^{18}O_R$ (light gray), $\Delta^{13}C_R$ (dark gray) and tree-ring width (black) with GPP. **b** Correlations of soil and leaf water $\delta^{18}O$ components of $\delta^{18}O_R$ with GPP. For Annecy the period of analysis is 1971–2000. Significance at the 95% confidence level is indicated by the black dots.