# Peer review of "A triple tree-ring constraint for tree growth and physiology in a global land surface model"

_Biogeosciences, 2020_

## Referee Comment (RC1) · Anonymous Referee #1 · 6 Jan 2021

Review of: "A triple tree-ring constraint for tree growth and physiology in a global land surface model"

This paper describes the parameterization and testing of a quasi-mechanistic large-scale model of forest growth. It tests the model against two other models and against measured data. The novelty of the manuscript lies in these model tests, especially because they include stable isotopic data, which both illuminate the physiological processes that cause the growth differences and provide tests of the mechanistic basis of the model. The significance is that this attempts to link a global-scale land-surface model to three kinds of tree-ring data. If the model performs well, it may be justified

to use it to describe the long tree-ring time series–potentially well beyond the range of remotely sensed, or even instrumental data.

The paper is mostly well-written, clearly significant, and appropriate for this journal. I particularly enjoyed reading the introduction and the methods and materials, which provide access to this subject for a broad audience. The analysis represents a tremendous breadth of work. I heartily applaud the authors for building these isotopic tests into their models and appreciate the comparisons to other isotopically enabled models—and to measured data. However, apart from the Intro and Methods, I found the paper difficult to read. There is so much here that the emphasis gets lost.

The abstract for a paper this complex should provide a roadmap that leads the reader to the main conclusion. It should mention not only ORCHIDEE, but also the other models, with a bit of explanation of why they were included. Also, Figure 1d-f is presented as a visual test of the models. If so, my visual impression is that MAIDENiso fails as the response surface looks quite different from that of the observations. This result should appear in the abstract. I suggest a change in the emphasis of the manuscript below. If accepted, this change should be reflected in the abstract.

MAIDENiso is referred to as "specialized" in at least two places in the manuscript. The model is described briefly on page 3 L4-8, but I was left wishing for a clearer description of what makes it different. Like many of your readers, I have never used it. This will be especially important if you choose to emphasize Fig. 1d-f.

The simulated results are not always distinguished clearly from the empirical data. This is especially important because you are comparing the models to empirical isotopic data. In particular: 1/10-13: I presume all the "physiological" data here are simulated? If so, say so, especially in the abstract. Have there been any direct measurements of, e.g., GPP at the Fontainebleau site? The same question arises about source water below. As these are all simulated, they should be labelled as such (e.g., 13/6-9).

The LPX-Bern results are barely mentioned in the text and the only conclusion they

lead to is that the model has "better isotopic forcing." What does that mean? Does LPX-Bern use different algorithms to estimate source water and water vapour? If so, it would be interesting to see how the predictions compare. The fact that the LPX-Bern model works better than either of the others for ïĄd'18O dilutes the impact of the presentation of ORCHIDEE. I suggest, to create a clearer emphasis in the paper, to either move LPX-Bern to a supplement or to discuss it in more detail. A particularly interesting detail would be a discussion of what might be changed in future versions of ORCHIDEE and MAIDENiso to make them work as well.

Section 3.1.2: I'm not sure I understand the purpose of this long section although I've read it several times. I think it is being presented as a test of the relationships embedded in the model structures and parameterizations. If so, this seems important and the isotopic methods seem ideally suited to it. I would make this the main emphasis of the paper. However, I noted that the MAIDENiso response surface looks really different from ORCHIDEE and from the data in Fig 1 d-f. I did not find this described clearly in the text. There was some description of the r-values of the partial correlations, but it is the slopes that catch the eye. The slope differences result in very different geometries across the response surfaces and this is what I would emphasize. Please note that the presentation of the response surfaces was interrupted by inferences about temperature and stomatal conductance, which I would move to the discussion. This section should end with a general model evaluation that addresses the visual impression that MAIDENiso has a problem.

The manuscript also describes isotopic changes in response to climate change and CO2. Although this is an interesting application of the model, it seems to belong in another paper. This impression is strengthened by the fact that the analysis neglects recent discussion of the effect of height growth on isotope ratios (and presumably growth)(Brienen et al., 2017; Marchand et al., 2020; Marshall & Monserud, 1996, 2006; Voelker et al., 2016). If it is to remain, the height issue must be addressed and information about height growth in these trees should be added. Are these trees are still

young enough to be growing in height? How tall were they? It would be great to see these height effects added to some future version of the model!

The interpretation of tree-ring 18O data is notoriously difficult and the Scheidegger et al. approach, although clever, is too simplistic. Because the authors cite Roden and Siegwolf (2012) (19/13-19), I presume that they appreciate the difficulty, but they do not express it in a way that a naïve reader is likely to detect. I suggest clearly and bluntly recognizing these difficulties for the people who will follow down this path. Related to this problem is the question of how the source water and water vapour 18O were simulated for this analysis. It should be described, at least briefly. The results are contingent on how this was done and how well it worked. This is necessary in part because the source water data are emphasized, e.g., in Figs. 2 and 6.

The temporal autocorrelation and its likely causes are interesting and important, but inadequately described. I would like to see a more carefully approach to this. In particular, there are mechanisms besides photosynthate carryover that could cause it. These include, for example, root or leaf mortality or production that might influence hydraulic balance in subsequent years. Monserud and Marshall speculate on some of these (2001). Whatever the mechanism, it would be great to have these effects described by the model and I support the emphasis placed on it.

It would be unfortunate if the main points of this manuscript were missed or misunderstood because of the complexity of presentation. I urge the authors to emphasize the response-surface tests of the models. If so, they might also expand the discussion of LPX-Bern and its better performance, including a comparison of the source and vapour 18O simulations. I suggest dropping the climate-change analysis for now. Especially if the the height effect were included in the model, the results would be significant enough to stand alone in another manuscript. Removing them from the current one would allow the model performance results to emerge clearly.

Brienen, R. J. W., Gloor, E., Clerici, S., Newton, R., Arppe, L., Boom, A., Bottrell, S.,

Callaghan, M., Heaton, T., Helama, S., Helle, G., Leng, M. J., Mielikäinen, K., Oinonen, M., & Timonen, M. (2017). Tree height strongly affects estimates of water-use efficiency responses to climate and CO 2 using isotopes. Nature Communications, 8(1), 288. https://doi.org/10.1038/s41467-017-00225-z Marchand, W., Girardin, M. P., Hartmann, H., Depardieu, C., Isabel, N., Gauthier, S., Boucher, É., & Bergeron, Y. (2020). Strong overestimation of water-use efficiency responses to rising CO2 in tree-ring studies. Global Change Biology, 26(8), 4538–4558. https://doi.org/10.1111/gcb.15166 Marshall, J. D., & Monserud, R. A. (1996). Homeostatic gas-exchange parameters inferred from 13C/12C in tree rings of conifers. Oecologia, 105(1), 13–21. Marshall, J. D., & Monserud, R. A. (2006). Co-occurring species differ in tree-ring $\delta$18O trends. Tree Physiology, 26(8), 1055–1066. Monserud, R. A., & Marshall, J. D. (2001). Time-series analysis of $\delta$13C from tree rings. I. Time trends and autocorrelation. Tree Physiology, 21(15), 1087–1102. Voelker, S. L., Brooks, J. R., Meinzer, F. C., Anderson, R., Bader, M. K.-F., Battipaglia, G., Becklin, K. M., Beerling, D., Bert, D., Betancourt, J. L., Dawson, T. E., Domec, J.-C., Guyette, R. P., Körner, C., Leavitt, S. W., Linder, S., Marshall, J. D., Mildner, M., Ogée, J., . . . Wingate, L. (2016). A dynamic leaf gas-exchange strategy is conserved in woody plants under changing ambient CO2: Evidence from carbon isotope discrimination in paleo and CO2 enrichment studies. Global Change Biology, 22(2), 889–902. https://doi.org/10.1111/gcb.13102

---

## Referee Comment (RC2) · Anonymous Referee #2 · 11 Jan 2021

In the present study, Barichivich and co-authors explore processes and historical changes of tree growth and tree physiology in a land surface model by simulating three tree ring proxies (namely ring width, carbon and oxygen isotopes) and by comparing them to observations in temperate and boreal sites (one specific site in Fontainebleau and a network of 5 other sites encompassing deciduous and conifer tree species). Further, the land surface model performance is compared to two other models at site and network level. Such approach and evaluation of Land surface model for long term tree growth and tree physiology variability is relevant and will certainly contribute to the understanding of carbon uptake and evapotranspiration dynamics in forested ecosystems; and will improve the predictive skills of tree/forest growth and carbon water cycles

responses to projected environmental changes.

The author conduct thorough simulations and analyses and the study is well designed. There are a few major points that can be addressed or explained better to clarify the results and their implications and highlight the relevance of the study presented here.

1- The introduction can be refocused into the potential of existing tree ring data to evaluate LSM and why is such work relevant to specific global change questions. The authors mention that but never make the case for it. What knowledge will be gained in term of processes by simulating tree ring attributes and comparing them to observations and output of other models. How do the three models differ which will contextualize the results and the discussion of their performance, specifically ORCHIDEE which is the major one being evaluated. 2- The results can be structured to better follow the study design. Site level (Fontainebleau) comparison of ORCHIDEE, MAINDENISO and observations and then the other sites where LPX-Bern model outputs are also used to compare with observations and ORCHIDEE. LPX-Bern is briefly described and then appears again in the discussion. In this regard the methods can clarify the forcing of all three models. 3- The Discussion relies heavily on descriptive results and does not highlight the physiological processes (beyond the use of carbohydrates and even so, this point needs more careful consideration) that can potentially explain the model-data comparison (or mismatch). In this regard, uncertainties in of tree ring proxies and modeling assumptions (iWUE Farquhar model, leaf water enrichment model, source water $\delta$18O forcing) are not addressed or discussed. 4- The references can be more updated in terms of recent efforts in using tree rings to benchmark process-based models but also to reflect the appropriate papers describing the mechanistic links between tree physiology and isotope variations in tree rings (specifically the O isotopes, beyond the review of McCaroll and Loader 20004).

Detailed Comments:

Pg1, Line 20: Their responses to what? Increasing atmospheric CO2, changing climate, disturbances?

Pg2, line 12: A suggestion would be to change adapt and perish as follows: how trees perish or adapt to environmental change is still limited.

Pg2, lines 14-15: additional references are relevant here specifically when using tree rings to either parametrize or evaluate mechanistic physiological models: • Lavergne, A. et al. Modelling tree ring cellulose $\delta$18O variations in two temperature-sensitive tree species from North and South America. Clim. Past 13, 1515–1526 (2017). • Belmecheri, S., Wright, W. E., Szejner, P., Morino, K. A. & Monson, R. K. Carbon and oxygen isotope fractionations in tree rings reveal interactions between cambial phenology and seasonal climate. Plant. Cell Environ. (2018). • Lavergne, A. et al. Historical changes in the stomatal limitation of photosynthesis: empirical support for an optimality principle. New Phytol. 225, 2484–2497 (2020).

Pg2, lines 19-20: These references correspond mostly to mature trees exposed to elevated CO2. The present study investigate historical records and model simulations of tree response to gradual increase of atmospheric CO2. As such, this ought to be highlighted as well.

Pg2, lines 25-30. This statement is misleading. Using a concept such as "cursed" imply an inherent unsuitability of ring width proxy for growth reconstructions. This is not true if the sampling strategy is adequately designed for that purpose. Indeed, the ITRDB repository includes trees collected mainly for climate reconstructions and it is well known that when using the same data for inferences of growth and specifically productivity, the data will reflect the growth dynamics and sensitivities of old, mature, climate sensitive individuals. It is not clear what is the point being made by the authors here? Why not test then model assumption based on collection specifically made for growth/productivity reconstructions? There are a few existing records (ecological sampling methods applied in Flux tower sites for e.g.). There is a great potential to tap tree ring data to benchmark LSM.

[Figure]

The introduction can make a stronger case for the use of both ORCHIDEE and MAIDEN iso. Why compare both models and what information or improvements can be gained from using then ORCHIDEE.

P4, Line 25, it is not clear whether the soil hydrology was modeled using an older version compared to the multi-layer cited after. If so, what is the motivation for this choice. Otherwise, it unnecessary to cite/describe what is not used.

P7 line 18, Where does the assumption of the effective path length of 8 mm comes from? How is this universally applied to different tree species/locations? See Roden et al. 2015,  c Roden, J., Kahmen, A., Buchmann, N. & Siegwolf, R. The enigma of effective path length for 18O enrichment in leaf water of conifers. Plant. Cell Environ. 38, 2551–2565 (2015). P11, lines 9-13. Why was this approach used to evaluate the relative contribution of source water versus evaporative enrichment, this is a statistical inference and will not reflect the mechanistic relationship between cellulose and leaf/source water $\delta$18O. For tree ring observations, a more adequate test would be using a proxy forward model (Evans et al., 2006) to evaluate how recorded $\delta$18O in tree ring cellulose compares to the modeled one using input of source water from observations (when available) or from the LMDz; and how varying source water and relative humidity (or VPD) affect the results. In the model world, these parameters are also used to simulate tree ring $\delta$18O and could similarly be compared first to the observations in order to evaluate the relative role of source versus leaf water evaporative enrichment.  c Evans, M. N. et al. A forward modeling approach to paleoclimatic interpretation of tree-ring data. J. Geophys. Res. Biogeosciences 111, (2006).

This also brings the point of potential uncertainty related to the assumptions of the péclet effect and the fraction of oxygen atom during cellulose synthesis. These were shown to vary along aridity gradients and intra-seasonally (Cheesman and Cernusak 2016) and with cell-size (lumen area, Szejner et al., 2020).  c Szejner, P., Clute, T., Anderson, E., Evans, M. N. & Hu, J. Reduction in lumen area is associated with the $\delta$18O exchange between sugars and source water during cellulose synthesis. New

Phytol. 226, 1583–1593 (2020). • Cheesman, A. W. & Cernusak, L. A. Infidelity in the outback: climate signal recorded in $\Delta$ 18 O of leaf but not branch cellulose of eucalypts across an Australian aridity gradient. Tree Physiol. 37, 554–564 (2016).

Replace carrying over with carryover and specify when first mentioned that it is a carry-over of carbohydrates from previous year or season. This could be discussed in more detail: the dynamic of stored versus recently assimilated photosynthates throughout the growing season (conifer vs deciduous) and under climate extremes (droughts) or other disturbances.

In the results section, when comparing ORCHIDEE and Maiden iso models, it is not clear whether the input data for running both models were the same (e.g. Source water, meteorology, etc)?

P12, lines $\sim$ 25. The effect of stomatal conductance on isotopic discrimination should also be recorded in leaf water enrichment. It is obviously not the case since $\delta$18O does not show a linear relationship with ring width. How would you explain this decoupling of isotopic responses to a reduction in stomatal conductance?

Atmospheric CO2 and $\delta$13C data used for calculation and simulation of $\delta$13C should be revised to the most updated datasets. This is discussed in details in Belmecheri and Lavergne 2020 with suggested recommendation for historical data.

• Belmecheri, S. & Lavergne, A. Compiled records of atmospheric CO2 concentrations and stable carbon isotopes to reconstruct climate and derive plant ecophysiological indices from tree rings. Dendrochronologia 63, 125748 (2020).

P14, lines 12. What is the evaluation metric for "well simulated" The model simulate less than 40% of the observed variability with 70% of unexplained variance. How do the authors assess that the model performance is good? Figure 5 is a great visualization for model performance summary. The result section can rely on this Figure for a consistent description of model performance and model-data comparison. For

instance, "moderate" category is never cited in the result text.

P14, lines 14. How could the model parametrization be biased towards temperate and deciduous forests since PFTs are informed in the model and climate drivers should reflect the forest/tree growing conditions (knowing that Mainden iso was calibrated for Fontainebleau site, but for ORCHIDEE?). In this case, which parameters are thought to be biased towards temperate forests.

P14, lines 15-17. If autocorrelation is removed from observations, does it improve model/data comparison since ORCHIDEE simulates poorly carbohydrate carryover.

P14, lines 19-20. Could the amplitude discrepancy between observed and simulated $\delta$13C be explained by post photosynthetic fractionations?

The Farquhar model referenced in the paper and used in the present study describes the isotopic discrimination at the leaf level (See Frank et al. 2015). It is not clear from the methods that ORCHIDEE takes it into account nor that the measurements of tree rings were scaled to the leaf level.

• Frank, D. C. et al. Water-use efficiency and transpiration across European forests during the Anthropocene. Nat. Clim. Chang. 5, 579–583 (2015).

The sensitivity of the simulated tree ring $\delta$18O depends on the selected months of model outputs. In the methods, the authors describe selecting May-August. Was this informed by knowledge of the growing season? The authors stated that using this window ensures a standard time window to compare all sites and isotopes (although different time windows for C and O isotopes did show different levels of agreement between observed and simulated isotopic variations). While the choice of a standard time-window can be justified for the reason outlined by the authors, it is not clear that such justification is beneficial when there is a loss of statistical agreement between data/model and if this choice is not informed (even roughly) by the tree's growing season. An important consideration for cellulose $\delta$18O is the timing and duration of

cell-wall thickness during which most of the cellulose is deposited. This will determine the isotopic ratio recorded in tree rings and the time window not only vary by latitude/attitude but can be narrower than anticipated (See Cuny et al. 2015).

Cuny, H. E. et al. Woody biomass production lags stem-girth increase by over one month in coniferous forests. Nat. Plants 1, 15160 (2015). In addition, because a fraction of leaf water will exchange with xylem water, C isotopes will carry more signal of previous carbohydrates compared to O isotopes (dampened carryover signal).

Pg 15, lines 5-10. Intuitively, GPP is expected to correlate with D13C , yet it is not the case here and D18O correlating better with GPP is rather intriguing. What is the mechanistic link to explain evaporative enrichment correlation with GPP. If this is driven by stomatal conductance so should D13C.

Pg16, Lines 25. This can be easily tested by removing auto-correlation from observed TRW prior to comparison with modeled TRW. As a side note (not a criticism to the present study). RW simulation can be tested using the Vaganov–Shashkin (VS) model to simulate TRW and compare ORCHIDEE performance to the VS model (similar approach to comparing ORCHIDEE and MAINDEN iso). This may shed more light into the poor ORCHIDEE performance in reproducing high frequency TRW variability.

Pg16, Lines 35. This assumption of drought legacy recovery has been tested for triple-proxies such as this study and the outcome depends largely on the detrending methods used for TRW. It also depends on the frequency of droughts. Hence, the results obtained by ORCHIDEE simulation might not be due only to a poor performance of the model. • Szejner, P., Belmecheri, S., Ehleringer, J. R. & Monson, R. K. Recent increases in drought frequency cause observed multi-year drought legacies in the tree rings of semi-arid forests. Oecologia 192, 241–259 (2020). The authors make a case of the 1976 drought to discuss limitation of the processes represented in the various models. The impact and recovery from the 1976 drought could be further elaborated using Superposed Epoch Analyses on simulated and observed tree ring proxies.

[Figure]

Pg 17, lines 25-30. This still contrast with an increase of D13C documented in atmospheric CO2 (Keeling et al., 2017). There is no discussion about the discrepancy between continuous increase of simulated D13C and the "pause" of D13C tree ring measurements since the 1980.

• Keeling, R. F. et al. Atmospheric evidence for a global secular increase in carbon isotopic discrimination of land photosynthesis. Proc. Natl. Acad. Sci. 114, 10361–10366 (2017). P18, lines ∼5. Are the simulated historical trends of $\delta$18O consistent with other observations (paleoclimate studies) in other forests, climatic regions?

P18, lines ∼15. How about the mechanistic representation of source water vs leaf water enrichment contribution in LPX-Bern model? If the forcing for the three models are not the same (specifically for source water 18O), the comparison of model performances is then biased.

P18, lines 30. Could it be that the sensitivity of stomatal conductance to factors other than CO2 be misrepresented (soil moisture for example).

---

## Author Comment (AC1) · 22 Mar 2021

**Response to anonymous Referee #1**

**This paper describes the parameterization and testing of a quasi-mechanistic large-scale model of forest growth. It tests the model against two other models and against measured data. The novelty of the manuscript lies in these model tests, especially because they include stable isotopic data, which both illuminate the physiological processes that cause the growth differences and provide tests of the mechanistic basis of the model. The significance is that this attempts to link a global-scale land-surface model to three kinds of tree-ring data. If the model performs well, it may be justified to use it to describe the long tree-ring time series–potentially well beyond the range of remotely sensed, or even instrumental data.**

**The paper is mostly well-written, clearly significant, and appropriate for this journal. I particularly enjoyed reading the introduction and the methods and materials, which provide access to this subject for a broad audience. The analysis represents a tremendous breadth of work. I heartily applaud the authors for building these isotopic tests into their models and appreciate the comparisons to other isotopically enabled models and to measured data. However, apart from the Intro and Methods, I found the paper difficult to read. There is so much here that the emphasis gets lost.**

We thank the reviewer for recognising the scope and magnitude of our work and for constructively pointing to areas for improvement. As we explain below, we have taken on board the suggestions to improve the focus and highlight the key elements of our triple tree-ring constraint.

**The abstract for a paper this complex should provide a roadmap that leads the reader to the main conclusion. It should mention not only ORCHIDEE, but also the other models, with a bit of explanation of why they were included. Also, Figure 1d-f is presented as a visual test of the models. If so, my visual impression is that MAIDENiso fails as the response surface looks quite different from that of the observations. This result should appear in the abstract. I suggest a change in the emphasis of the manuscript below. If accepted, this change should be reflected in the abstract.**

We followed the advice of the reviewer regarding the improvement of the emphasis of the paper. In the revised abstract we now mention the other models as a brief model context for the performance of ORCHIDEE but we do not develop in their evaluation to avoid losing focus on ORCHIDEE. The poor performance of MAIDENiso is discussed in the new Discussion subsection 4.2 at the end of the model evaluation with the tree-ring triplet.

**MAIDENiso is referred to as "specialized" in at least two places in the manuscript. The model is described briefly on page 3 L4-8, but I was left wishing for a clearer description of what makes it different. Like many of your readers, I have never used it. This will be especially important if you choose to emphasize Fig. 1d-f.**

The introduction is now more streamlined to highlight how the main growth processes are represented in current modelling approaches. It is hopefully much more clear where each model stands in the model landscape. We also added a compact description of the setup for the published simulations and processes included in MAIDENiso and LPX-Bern in the last two paragraphs of section 2.3 (Simulations)

**The simulated results are not always distinguished clearly from the empirical data. This is especially important because you are comparing the models to empirical isotopic data. In particular: 1/10-13: I presume all the "physiological" data here are simulated? If so, say so, especially in the abstract. Have there been any direct measurements of, e.g., GPP at the Fontainebleau site? The same question arises about source water below. As these are all simulated, they should be labelled as such (e.g., 13/6-9).**
We revised the text to make a clear distinction of when the description is based on simulated or observed data. The GPP referred to in the manuscript is simulated by ORCHIDEE. We did not use the short eddy-covariance measurements available for a forest near Fontainebleau in this study. Similarly, all source water is simulated and is now explicitly labelled as such.

**The LPX-Bern results are barely mentioned in the text and the only conclusion they lead to is that the model has "better isotopic forcing." What does that mean? Does LPX-Bern use different algorithms to estimate source water and water vapour? If so, it would be interesting to see how the predictions compare. The fact that the LPX-Bern model works better than either of the others for ïA˛d'18O dilutes the impact of the presentation of ORCHIDEE. I suggest, to create a clearer emphasis in the paper, to either move LPX-Bern to a supplement or to discuss it in more detail. A particularly interesting detail would be a discussion of what might be changed in future versions of ORCHIDEE and MAIDENiso to make them work as well.**
We followed the advice of the reviewer to keep the focus on the tree-ring triplet and deleted the former Figures 2-4 (the triplet with only leaf water enrichment, climate response and water use efficiency). In the revised manuscript, the former Figure 5 is now Figure 2 (Taylor diagram of model performance). This gave us space for the new Figure 3 to analyse more deeply the behaviour of LPX-Bern versus ORCHIDEE with respect to d18O. The new Figure 3 shows a detailed comparison of the contributions of source water and leaf water enrichment to the simulated d18O signals simulated by both models and also the correlation of each component of the forcing (d18O of precipitation, d18O of vapour and d18O of source water) and the resulting simulations of d18O in leaf water and cellulose. This Figure allows disentangling and comparing the nature of the simulated signals in both models and evaluating the consistency of the isotopic drivers used to force the models (i.e., d18O of precipitation and d18O of vapour from the LMDZ atmospheric for ORCHIDEE and d18O of source water (i.e. soil water) from the ECHAM5 atmospheric model for LPX-Bern).

The critical process and areas of improvement identified in the evaluation of ORCHIDEE against tree-ring data are now more clearly discussed in Section 4.1 and 4.2. The casue of the issue of MAIDENiso is addressed in Section 4.2, but we refrained to develop detailed recomendations for this model since it was used only for comparison purposes. However we make clear that the cause of the poorly simulated growth-isotope surface is predominantly

due to the excessive carryover in simulated ring width, which decouples growth from leaf-level responses.

**Section 3.1.2: I'm not sure I understand the purpose of this long section although I've read it several times. I think it is being presented as a test of the relationships embedded in the model structures and parameterizations. If so, this seems important and the isotopic methods seem ideally suited to it. I would make this the main emphasis of the paper.**

Following the advice of the reviewer, the narrative flow of section 3.1.2 was revised and the section was renamed as "Simulated tree-ring triplet" to emphasize the triple tree-ring constraint. Former sections 3.1.3 (climate response) and 3.1.4 (20th century water use efficiency) were deleted to improve the focus of the paper.

**However, I noted that the MAIDENiso response surface looks really different from ORCHIDEE and from the data in Fig 1 d-f. I did not find this described clearly in the text. There was some description of the r-values of the partial correlations, but it is the slopes that catch the eye. The slope differences result in very different geometries across the response surfaces and this is what I would emphasize. Please note that the presentation of the response surfaces was interrupted by inferences about temperature and stomatal conductance, which I would move to the discussion. This section should end with a general model evaluation that addresses the visual impression that MAIDENiso has a problem.**

We thank the reviewer for pointing our attention to the slopes of the surface. Indeed, the simple regression slopes better quantify the geometry of the triplet and the coupling of the processes that it represents. We now labeled the regression slopes in Fig 1 from b1 to b3 and interpreted their magnitude in terms of processes. Since these inter-relationships are the basis of the novel tree-ring triplet presented in the paper we discussed the meaning of the slopes and relationships in terms of processes in the new section 4.2, which integrates the former sections 4.1 (Integrating tree-ring and carbon...) and 4.3 (Constraining model processes..).

**The manuscript also describes isotopic changes in response to climate change and CO2. Although this is an interesting application of the model, it seems to belong in an- other paper. This impression is strengthened by the fact that the analysis neglects recent discussion of the effect of height growth on isotope ratios (and presumably growth) (Brienen et al., 2017; Marchand et al., 2020; Marshall & Monserud, 1996, 2006; Voelker et al., 2016). If it is to remain, the height issue must be addressed and infor- mation about height growth in these trees should be added. Are these trees are still young enough to be growing in height? How tall were they? It would be great to see these height effects added to some future version of the model!**

Following the advice of the reviewer, we decided to remove the analysis of 20th century iWUE and publish it as a follow up letter. The effect of tree height on carbon isotopes is not explicitly represented in ORCHIDEE. This has been shown to be particularly important in deep canopy tropical forests and certainly can account for initial trends during the juvenile period. The age of the stand in Fontainebleau is about 120 years (given in Table 1) and tree height is between 20-25 m. A newer version of ORCHIDEE introduced a better representation of forest structure and light penetration

(https://gmd.copernicus.org/preprints/gmd-2020-29/) and could be used to address this issue, particularly for dense tropical forests.

**The interpretation of tree-ring d18O data is notoriously difficult and the Scheidegger et al. approach, although clever, is too simplistic. Because the authors cite Roden and Siegwolf (2012) (19/13-19), I presume that they appreciate the difficulty, but they do not express it in a way that a naïve reader is likely to detect. I suggest clearly and bluntly recognizing these difficulties for the people who will follow down this path. Related to this problem is the question of how the source water and water vapour d18O were simulated for this analysis. It should be described, at least briefly. The results are contingent on how this was done and how well it worked. This is necessary in part because the source water data are emphasized, e.g., in Figs. 2 and 6.**

The uncertainties and critical processes identified for the simulation of the tree-ring variables are now explicitly addressed at the beginning of the Discussion in the new Section 4.1. We explicitly recognise the difficulty for simulting d18O in the 5th paragraph.

As mentioned earlier, the external d18O isotopic forcings used for each model (ORCHIDEE, MAIDENiso, LPX-Bern) are now better described in Section 2.3 (simulations). The new Figure 3 also helps to understand the origin of the differences and similarities between tree-ring dO18 simulated by ORCHIDEE and LPX-Bern.

**The temporal autocorrelation and its likely causes are interesting and important, but inadequately described. I would like to see a more carefully approach to this. In particular, there are mechanisms besides photosynthate carryover that could cause it. These include, for example, root or leaf mortality or production that might influence hydraulic balance in subsequent years. Monserud and Marshall speculate on some of these (2001). Whatever the mechanism, it would be great to have these effects described by the model and I support the emphasis placed on it.**

Since the carryover is one of the critical processes to simulate tree-ring width variability, we further developed the discussion of its causal factors in the first paragraph of section 4.1 of the Discussion. Factors other than carbohydrate remobilization are now mentioned and referenced.

**It would be unfortunate if the main points of this manuscript were missed or misunderstood because of the complexity of presentation. I urge the authors to emphasize the response-surface tests of the models. If so, they might also expand the discussion of LPX-Bern and its better performance, including a comparison of the source and vapour d18O simulations. I suggest dropping the climate-change analysis for now. Especially if the height effect were included in the model, the results would be significant enough to stand alone in another manuscript. Removing them from the current one would allow the model performance results to emerge clearly.**

We thank the reviewer for all the constructive advice, which we followed in order to improve the focus and presentation of the paper. The revised manuscript highlights the response surface of the tree-ring triplet and the comparison of ORCHIDEE with LPX-Bern as global model benchmark, including a concise analysis of the isotopic forcings.

Brienen, R. J. W., Gloor, E., Clerici, S., Newton, R., Arppe, L., Boom, A., Bottrell, S., Callaghan, M., Heaton, T., Helama, S., Helle, G., Leng, M. J., Mielikäinen, K., Oinonen, M., & Timonen, M. (2017). Tree height strongly affects estimates of water-use efficiency responses to climate and CO 2 using isotopes. Nature Communications, 8(1), 288. https://doi.org/10.1038/s41467-017-00225-z Marchand, W., Girardin, M. P., Hartmann, H., Depardieu, C., Isabel, N., Gauthier, S., Boucher, É., & Bergeron, Y. (2020). Strong overestimation of water-use efficiency responses to rising CO2 in tree-ring stud- ies. Global Change Biology, 26(8), 4538–4558. https://doi.org/10.1111/gcb.15166 Mar- shall, J. D., & Monserud, R. A. (1996). Homeostatic gas-exchange parameters inferred from 13C/12C in tree rings of conifers. Oecologia, 105(1), 13–21. Marshall, J. D., & Monserud, R. A. (2006). Co-occurring species differ in tree-ring $\delta$18O trends. Tree Physiology, 26(8), 1055–1066. Monserud, R. A., & Marshall, J. D. (2001). Time-series analysis of $\delta$13C from tree rings. I. Time trends and autocorrelation. Tree Physiology, 21(15), 1087–1102. Voelker, S. L., Brooks, J. R., Meinzer, F. C., Anderson, R., Bader,M. K.-F., Battipaglia, G., Becklin, K. M., Beerling, D., Bert, D., Betancourt, J. L., Daw- son, T. E., Domec, J.-C., Guyette, R. P., Körner, C., Leavitt, S. W., Linder, S., Marshall, J. D., Mildner, M., Ogée, J., . . . Wingate, L. (2016). A dynamic leaf gas-exchange strategy is conserved in woody plants under changing ambient CO2: Evidence from carbon isotope discrimination in paleo and CO2 enrichment studies. Global Change Biology, 22(2), 889–902.

---

## Author Comment (AC2) · 22 Mar 2021

**Response to anonymous Referee #2**

**In the present study, Barichivich and co-authors explore processes and historical changes of tree growth and tree physiology in a land surface model by simulating three tree ring proxies (namely ring width, carbon and oxygen isotopes) and by comparing them to observations in temperate and boreal sites (one specific site in Fontainebleau and a network of 5 other sites encompassing deciduous and conifer tree species). Further, the land surface model performance is compared to two other models at site and network level. Such approach and evaluation of Land surface model for long term tree growth and tree physiology variability is relevant and will certainly contribute to the understanding of carbon uptake and evapotranspiration dynamics in forested ecosystems; and will improve the predictive skills of tree/forest growth and carbon water cycles responses to projected environmental changes.**

**The authors conduct thorough simulations and analyses and the study is well designed. There are a few major points that can be addressed or explained better to clarify the results and their implications and highlight the relevance of the study presented here.**

**1. The introduction can be refocused into the potential of existing tree ring data to evaluate LSM and why is such work relevant to specific global change questions. The authors mention that but never make the case for it. What knowledge will be gained in term of processes by simulating tree ring attributes and comparing them to observations and output of other models. How do the three models differ which will contextualize the results and the discussion of their performance, specifically ORCHIDEE which is the major one being evaluated.**

We thank the reviewer for the constructive advice. Following this comment we have revised and extensively rewritten the introduction to better explain the need for improvement of land surface models and why tree rings are especially suited for doing it at long timescales. The different modelling approaches are briefly explained and provide the context for process descriptions and the use of MAIDENiso and LPX-Bern for model-to-model comparisons.

**2. The results can be structured to better follow the study design. Site level (Fontainebleau) comparison of ORCHIDEE, MAINDENISO and observations and then the other sites where LPX-Bern model outputs are also used to compare with observations and ORCHIDEE. LPX-Bern is briefly described and then appears again in the discussion. In this regard the methods can clarify the forcing of all three models.**

We followed the advice of the two reviewers regarding the focus of the paper and organisation of the results and discussion sections of the manuscript. The new Figure 3 addresses the detailed comparison between ORCHIDEE and LPX-Bern and the external d18O isotopic forcings used for each model (ORCHIDEE, MAIDENiso, LPX-Bern) are now better described in Section 2.3 (simulations).

**3. The Discussion relies heavily on descriptive results and does not highlight the physiological processes (beyond the use of carbohydrates and even so, this point needs more careful consideration) that can potentially explain the model-data comparison (or mismatch). In this regard, uncertainties in of tree ring proxies and modeling assumptions (iWUE Farquhar model, leaf water enrichment model, source water $\delta 18O$ forcing) are not addressed or discussed.**

The discussion was rewritten to recognise uncertainties and highlight the physiological interpretation of tree-ring data and the unique mechanistic value of combining growth and isotopes in the tree-ring triplet to constrain models. The uncertainties for each variable are explicitly discussed in section 4.1 (Uncertainties and critical processes…) and the physiological interpretation of the tree-ring triplet is discussed in section 4.2 (Constraining model processes with the growth-isotope tree-ring triplet).

**4. The references can be more updated in terms of recent efforts in using tree rings to benchmark process-based models but also to reflect the appropriate papers describing the mechanistic links between tree physiology and isotope variations in tree rings (specifically the O isotopes, beyond the review of McCaroll and Loader 2004).**

The revised introduction and discussion incorporate more updated essential references, some of which were suggested by the reviewer.

Detailed Comments:
**Pg1, Line 20: Their responses to what? Increasing atmospheric CO2, changing climate, disturbances?**

We revised the clause for clarity and added "…their simulated responses to environmental changes…"

**Pg2, line 12: A suggestion would be to change adapt and perish as follows: how trees perish or adapt to environmental change is still limited.**

Suggestion taken.

**Pg2, lines 14-15: additional references are relevant here specifically when using tree rings to either parametrize or evaluate mechanistic physiological models:**
 **- Lavergne, A. et al. Modelling tree ring cellulose $\delta 18O$ variations in two temperature-sensitive tree species from North and South America. Clim. Past 13, 1515–1526 (2017).**
**- Belmecheri, S., Wright, W. E., Szejner, P., Morino, K. A. & Monson, R.K. Carbon and oxygen isotope fractionations in tree rings reveal interactions between cambial phenology and seasonal climate. Plant. Cell Environ. (2018).**
**- Lavergne, A. et al. Historical changes in the stomatal limitation of photosynthesis: empirical sup- port for an optimality principle. New Phytol. 225, 2484–2497 (2020).**

In this part we refer to some relevant references for empirical studies using tree-rings to infer ecological changes. Their application for modelling is introduced later. We added the reference of Lavergne et al 2020.

**Pg2, lines 19-20: These references correspond mostly to mature trees exposed to elevated CO2. The present study investigate historical records and model simulations of tree response to gradual increase of atmospheric CO2. As such, this ought to be highlighted as well.**

To reflect this point of the reviewer, the sentence was revised as follows "…to study the range of historical responses of mature trees to gradual global change or manipulative experiments"…

**Pg2, lines 25-30. This statement is misleading. Using a concept such as "cursed" imply an inherent unsuitability of ring width proxy for growth reconstructions. This is not true if the sampling strategy is adequately designed for that purpose. Indeed, the ITRDB repository includes trees collected mainly for climate reconstructions and it is well known that when using the same data for inferences of growth and specifically productivity, the data will reflect the growth dynamics and sensitivies of old, mature, climate sensitive individuals. It is not clear what is the point being made by the authors here? Why not test then model assumption based on collection specifically made for growth/productivity reconstructions? There are a few existing records (ecological sampling methods applied in Flux tower sites for e.g.).**

This paragraph introduces tree rings as a means to infer long-term changes in growth and physiology, highlighting their advantages and potential pitfalls. The magnitude and reach of the biases is still a contentious issue in the community as is reflected in this part of the text. The text was revised to better reflect that the issue applies mainly to the tree-ring data archived in the ITRDB and not inherently to all tree-ring data. We added a closing sentence stating that "The effect of sampling biases on long-term growth trends can be effectively minimized by using appropriated sampling designs for productivity reconstructions (Gough et al., 2008; Nehrbass-Ahles et al., 2014; Dye et al., 2016). "

**There is a great potential to tap tree ring data to benchmark LSM. The introduction can make a stronger case for the use of both ORCHIDEE and MAIDEN iso. Why compare both models and what information or improvements can be gained from using then ORCHIDEE.**

We agree. This is the aim of the paper. We revised the introduction and now we better explain why these models were used to provide a wider context for the few tree-ring enabled models.

**P4, Line 25, it is not clear whether the soil hydrology was modeled using an older version compared to the multi-layer cited after. If so, what is the motivation for this choice. Otherwise, it unnecessary to cite/describe what is not used.**

Point taken. The mention to the new multi-layer soil scheme was deleted.

**P7 line 18, Where does the assumption of the effective path length of 8 mm comes from? How is this universally applied to different tree species/locations? See Roden et al. 2015, - Roden, J., Kahmen, A., Buchmann, N. & Siegwolf, R. The enigma of effective path length for 18O enrichment in leaf water of conifers. Plant. Cell Environ. 38, 2551–2565 (2015).**

Thanks for pointing out this important source of uncertainty for simulating dO18 in global models limited to a few forest PFTs. LPX-Bern and ORCHIDEE have a similar representation of the Peclet effect and isotopic mixing during cellulose production but the Peclet parameterization differ with respect to the highly uncertain and variable L (path length). A value of 3 mm for all PFTs was used in LPX-Bern, which is almost three times lower than our L value. Our value of 8 mm was obtained by Risi et al., (2016) from tuning the model against seasonal isotopic observations in a few mid-latitude sites in Europe. The

impact of this high parameter value in ORCHIDEE is evident when compared with LPX-Bern because ORCHIDEE simulates a much stronger imprint of source water and dampening of the leaf enrichment signal in tree-ring d18O. This source of uncertainty and its consequences are discussed in the 6[th] paragraph of section 4.1.

**P11, lines 9-13. Why was this approach used to evaluate the relative contribution of source water versus evaporative enrichment, this is a statistical inference and will not reflect the mechanistic relationship between cellulose and leaf/source water δ18O. For tree ring observations, a more adequate test would be using a proxy forward model (Evans et al., 2006) to evaluate how recorded δ18O in tree ring cellulose compares to the modeled one using input of source water from observations (when available) or from the LMDz; and how varying source water and relative humidity (or VPD) affect the results. In the model world, these parameters are also used to simulate tree ring δ18O and could similarly be compared first to the observations in order to evaluate the relative role of source versus leaf water evaporative enrichment.**
**-Evans, M. N. et al. A forward modeling approach to paleoclimatic interpretation of tree-ring data. J. Geophys. Res. Biogeosciences 111, (2006).**
We opted for this statistical approach because it allows a simple partitioning of the Rsq due to source water and leaf water enrichment in simulated δ18O. This was not attempted for the observations because the real isotopic δ18O forcing in source water is unknown. We are aware of Evans' model, but adding another model to our comparison is out of scope.

**This also brings the point of potential uncertainty related to the assumptions of the péclet effect and the fraction of oxygen atom during cellulose synthesis. These were shown to vary along aridity gradients and intra-seasonally (Cheesman and Cernusak 2016) and with cell-size (lumen area, Szejner et al., 2020).**
**-Szejner, P., Clute, T., Anderson, E., Evans, M. N. & Hu, J. Reduction in lumen area is associated with the δ18O exchange between sugars and source water during cellulose synthesis. New Phytol. 226, 1583–1593 (2020).**
**-Cheesman, A. W. & Cernusak, L. A. Infidelity in the outback: climate signal recorded in Δ 18 O of leaf but not branch cellulose of eucalypts across an Australian aridity gradient. Tree Physiol. 37, 554–564 (2016).**
As explained above, these sources of uncertainty and their consequences for ORCHIDEE are now discussed in the 6[th] paragraph of section 4.1.

**Replace carrying over with carryover and specify when first mentioned that it is a carry- over of carbohydrates from previous year or season. This could be discussed in more detail: the dynamic of stored versus recently assimilated photosynthates throughout the growing season (conifer vs deciduous) and under climate extremes (droughts) or other disturbances.**
Thanks for pointing the mistake in the wording. It was corrected. The carryover and its causes is described in more depth than before in the first paragraph of section 4.1 in the Discussion.

**In the results section, when comparing ORCHIDEE and Maiden iso models, it is not clear whether the input data for running both models were the same (e.g. Source water, meteorology, etc)?**

The external d18O isotopic forcings for the already published simulations of MAIDENiso and LPX-Bern were different. This is is now better described in Section 2.3 (simulations).

**P12, lines 25. The effect of stomatal conductance on isotopic discrimination should also be recorded in leaf water enrichment. It is obviously not the case since δ18O does not show a linear relationship with ring width. How would you explain this decoupling of isotopic responses to a reduction in stomatal conductance?**

The dual carbon-oxygen isotope relationship is significant (r=-0.40, p<0.001), therefore stomatal conductance is recorded in leaf water enrichment. The non-linear and somewhat weaker relationship between ring width and oxygen variations points to a decoupling of growth and leaf responses, which arises from the growth memory present in the observations as growth would use a mixture of older and new carbon. This is discussed in the 3$^{rd}$ and 4$^{th}$ paragraphs of Section 4.1.

**Atmospheric CO2 and δ13C data used for calculation and simulation of δ13C should be revised to the most updated datasets. This is discussed in details in Belmecheri and Lavergne 2020 with suggested recommendation for historical data.**
-**Belmecheri, S. & Lavergne, A. Compiled records of atmospheric CO2 concentra-tions and stable carbon isotopes to reconstruct climate and derive plant ecophysiolog- ical indices from tree rings. Dendrochronologia 63, 125748 (2020).**

McCarroll and Loader (2004) dataset has long been the standard in the community. As Belmecheri and Lavergne pointed out, the main impact of the different sources occurs on the absolute values. Although we appreciate the suggestion of the reviewer, changing the CO2 and δ13Catm data would not change the interannual variability of carbon discrimination and iWUE, which is the main focus of our model-data comparisons. Therefore, we don't believe this change to be necessary given our focus on correlations.

**P14, lines 12. What is the evaluation metric for "well simulated" The model simulate less than 40% of the observed variability with 70% of unexplained variance. How do the authors assess that the model performance is good? Figure 5 is a great visualization for model performance summary. The result section can rely on this Figure for a consistent description of model performance and model-data comparison. For instance, "moderate" category is never cited in the result text.**

Although this is subjective and relative to the field, well simulated or well reconstructed in a tree-ring context is usually anything higher than 25-30% of the total variance. This corresponds to a correlation r>=0.5, which qualitatively in our Fig.2 (former Fig. 5) falls in the moderate category. We understand the point of the reviewer and we are more explicit with the qualitative adjectives of performance, linking them to the Fig. 2. We added the following sentence in the caption of the figure "Four qualitative areas of performance in terms of the magnitude of correlations or simulated variance are indicated as a visual aid"

**P14, lines 14. How could the model parametrization be biased towards temperate and deciduous forests since PFTs are informed in the model and climate drivers should reflect the forest/tree growing conditions (knowing that Maiden iso was calibrated for Fontainebleau site, but for ORCHIDEE?). In this case, which parameters are thought to be biased towards temperate forests.**

Given that this sentence is not extremely insightful and was not correctly understood by the referee it was deleted. This sentence was intended as a general concern and did not refer a specific process or a set of parameters. It expressed a general concern about ecological research and thus mechanistic modelling based on the outcomes of this ecological research. Much of our ecological understanding is based on research in temperate ecosystems, this understanding dominates textbooks and university courses and as such enters our land surface models. In the absence of knowledge specific to the boreal and tropical biomes the models fall back on our understanding (and parameters) of temperate ecosystems. This may have resulted in a bias of these models towards the temperate zone. A clear examples of the temperate bias can be found, for example, in how LSM formalize croplands (one rotation per year, homogeneous system, short growing seasons, fallow in between rotations) which represents intensive cropping in the temperate zone but does not represent at all cropping over most of Africa.

**P14, lines 15-17. If autocorrelation is removed from observations, does it improve model/data comparison since ORCHIDEE simulates poorly carbohydrate carryover.**
Doing prewhitening to the observations does improve slightly the model-data correlations in the French sites where autocorrelation is stronger. We have added two sentences about this possibility in the 1$^{st}$ paragraph of section 4.1 of the Discussion: "The carryover of ring-width observations can be removed through statistical prewhitening (Cook, 1985). A data-model comparison using prewhitened observations results in slightly higher correlations with simulated ring-width variability in the three French sites (e.g., increase in r from 0.50 to 0.56 in Fontainebleau) but not in the Finish sites (not shown), where autocorrelation was lower (Fig. 6a)."

**P14, lines 19-20. Could the amplitude discrepancy between observed and simulated δ13C be explained by post photosynthetic fractionations?**
**The Farquhar model referenced in the paper and used in the present study describes the isotopic discrimination at the leaf level (See Frank et al. 2015). It is not clear from the methods that ORCHIDEE takes it into account nor that the measurements of tree rings were scaled to the leaf level.**
**-Frank, D. C. et al. Water-use efficiency and transpiration across European forests during the Anthropocene. Nat. Clim. Chang. 5, 579–583 (2015).**
The caveats of using this simple formulation for carbon discrimination in land surface models were explicitly described in the last paragraph of section 2.1.2 in Methods. As the reviewer points out, it assumes that post-photosynthetic and mixing processes are negligible, which admittedly is a major simplification. However, as we mention it, it is still a useful model of carbon discrimination to constrain the environmental response of land surface models with tree-ring data as demonstrated in earlier studies (e.g., Bodin et al., 2013, Keel et al., 2016).

The possible causes for the underestimation of the interannual variability of carbon discrimination were also explicitly addressed in the former discussion section 4.2 L15-23. We stated that the amplitude of simulated carbon discrimination in Fontainebleau was very sensitive to soil depth and maximum photosynthetic capacity (Vcmax) parameters. Therefore, beside the possible effect of post-photosynthetic processes, the underestimation of the variability is in part due to these parameter values in the standard parameterization of the model PFTs. This is the message for the reader in the revised discussion (4$^{th}$ paragraph of section 4.1).

**The sensitivity of the simulated tree ring δ18O depends on the selected months of model outputs. In the methods, the authors describe selecting May-August. Was this informed by knowledge of the growing season? The authors stated that using this window ensures a standard time window to compare all sites and isotopes (although different time windows for C and O isotopes did show different levels of agreement between observed and simulated isotopic variations). While the choice of a standard time-window can be justified for the reason outlined by the authors, it is not clear that such justification is beneficial when there is a loss of statistical agreement between data/model and if this choice is not informed (even roughly) by the tree's growing season. An important consideration for cellulose δ18O is the timing and duration of cell-wall thickness during which most of the cellulose is deposited. This will determine the isotopic ratio recorded in tree rings and the time window not only vary by latitude/attitude but can be narrower than anticipated (See Cuny et al. 2015).**
**-Cuny, H. E. et al. Woody biomass production lags stem-girth increase by over one month in coniferous forests. Nat. Plants 1, 15160 (2015).**
**In addition, because a fraction of leaf water will exchange with xylem water, C isotopes will carry more signal of previous carbohydrates compared to O isotopes (dampened carryover signal).**

As we stated in the methods, the choice of a fixed window is a necessary compromise to ensure comparability between the isotopes across the sites. Although not explicitly said in the text, an exploration of different windows showed that a sufficiently wide window such as May-August better integrates the seasonal responses of simulated photosynthesis to soil moisture and temperature along the climate gradient, which are then comparable with the seasonally integrated observations. Shorter or longer seasons did not change substantially the correlations between simulated and observed carbon discrimination, but the oxygen isotope was sensitive to May. Michelot et al. (2011) showed that in Fontainebleau the growing season (photosynthesis) of oak starts in mid-April, reaching a peak in transpiration and photosynthesis in late-May to early-June, which coincides with the transition from earlywood to latewood. This helps explaining the sensitivity to May and also informed our choice for the May-August window. To address the concern of the reviewer we added the following sentence "Late-May and early-June correspond to the seasonal peak in transpiration and photosynthesis of oak in Fontainebleau, which is closely followed by the transition between earlywood and latewood (Michelot et al., 2011)."

Like other LSMs, ORCHIDEE (r898) does not consider the intraseasonal dynamics of non-structural carbohydrates. Thus, it does not capture the fine-scale temporal asynchrony between photosynthesis and carbon expenditure in wood growth and respiration as is seen in the observations (e.g., Cuny et al 2015). However, the wood formation process is highlighted in the conclusion as an area of improvement for the representation of tree growth in the model.

**Pg 15, lines 5-10. Intuitively, GPP is expected to correlate with D13C, yet it is not the case here and D18O correlating better with GPP is rather intriguing. What is the mechanistic link to explain evaporative enrichment correlation with GPP. If this is driven by stomatal conductance so should D13C.**

This point was addressed at the end of the Discussion in former section 4.4. Now it is addressed in the 1st and 2nd paragraph of section 4.3. D13C does correlate significantly with GPP as expected (shown in former Fig. 8 and now Fig. 6), but the correlation with of δ18O with GPP is more consistent spatially because of the synergistic effect between source water and leaf enrichment.

**Pg16, Lines 25. This can be easily tested by removing auto-correlation from observed TRW prior to comparison with modeled TRW. As a side note (not a criticism to the present study). RW simulation can be tested using the Vaganov–Shashkin (VS) model to simulate TRW and compare ORCHIDEE performance to the VS model (similar approach to comparing ORCHIDEE and MAINDEN iso). This may shed more light into the poor ORCHIDEE performance in reproducing high frequency TRW variability.**
We thank to the reviewer for pointing this out. As explained in the response above we added two sentences about the impact of using prewhitened observations in model-data comparison in the 1st paragraph of the Discussion.
We are very familiar with the use of the VS model and could have certainly been an alternative to MAIDENiso, however, it is based only on sink-driven growth and does not simulate stable isotopes. In addition, it would work only for the two coniferous sites. The semi-empirical alternative VS-Lite model applied to any species but is not process-based.

**Pg16, Lines 35. This assumption of drought legacy recovery has been tested for triple- proxies such as this study and the outcome depends largely on the detrending methods used for TRW. It also depends on the frequency of droughts. Hence, the results obtained by ORCHIDEE simulation might not be due only to a poor performance of the model.**
**- Szejner, P., Belmecheri, S., Ehleringer, J. R. & Monson, R. K. Recent in- creases in drought frequency cause observed multi-year drought legacies in the tree rings of semi-arid forests. Oecologia 192, 241–259 (2020).**
**The authors make a case of the 1976 drought to discuss limitation of the processes represented in the various models. The impact and recovery from the 1976 drought could be further elaborated using Superposed Epoch Analyses on simulated and observed tree ring proxies.**
SEA would be useful to statistically quantify the mean pattern of response to extremes in the model, but the result of such an analysis is obvious given the large differences in persistence between simulation and observations. It would also be redundant beside the autocorrelation analysis presented, which we believe illustrates the issue.

**Pg 17, lines 25-30. This still contrast with an increase of D13C documented in atmospheric CO2 (Keeling et al., 2017). There is no discussion about the discrepancy between continuous increase of simulated D13C and the "pause" of D13C tree ring measurements since the 1980.**
**Keeling, R. F. et al. Atmospheric evidence for a global secular increase in carbon isotopic discrimination of land photosynthesis. Proc. Natl. Acad. Sci. 114, 10361–10366 (2017).**
The analysis of water use efficiency was discarded following the advice of reviewer 1 to improve the focus of the paper on the tree-ring triplet. The ongoing reduction of the rate of increase in water use efficiency in recent decades has been documented in recent papers and will be the subject of a forthcoming paper with the model.

**P18, lines 5.  Are the simulated historical trends of _δ18O_ consistent with other observations (paleoclimate studies) in other forests, climatic regions?**

We only focused our analysis on the interannual variability of _δ18O_ and the analyses of trends is interesting but would be out of the scope, since it would add another dimension to the already long paper.

**P18, lines 15. How about the mechanistic representation of source water vs leaf water enrichment contribution in LPX-Bern model? If the forcing for the three models are not the same (specifically for source water 18O), the comparison of model performances is then biased.**

This is fully addressed in the new Fig. 3. Although the isotopic forcings of precipitation and vapour are different, the simulated isotopic signatures of source water and leaf water enrichment are significantly correlated between the two models because of the strong integration effect of soil water and the effect of the atmospheric forcing (VPD).

**P18, lines 30. Could it be that the sensitivity of stomatal conductance to factors other than CO2 be misrepresented (soil moisture for example).**

Yes. This is discussed in the 3[rd] and 4[th] paragraph of Section 4.2.

---

## Author Response (AR2)

**I wonder if the authors have considered whether the decision to analyze cellulose might also influence the ability of ORCHIDEE to simulate isotope composition of tree rings? The mechanistic modeling used in ORCHIDEE is not necessarily specific to cellulose content of trees.**
For carbon isotopes, we used the simple formulation of Farquhar. Admittedly, this is more comparable with leaf measurements than with cellulose in tree rings. However, this formulation has been commonly used in modeling studies because of its simplicity. This limitation is acknowledged in the Discussion (second paragraph of p16). Future studies will include a more complete parameterization for carbon isotopes that integrates better the post-photosynthetic and mixing processes that affect the carbon isotopic signatures of cellulose.

**Below are some other minor changes to consider, some of which are necessary: p. 3 line 20, "in this type of model"**
**p. 6 line 21, "represented following a similar formulation as in other isotope-enabled" line 22, "isotopic composition"**
**p. 10 line 25, "conducted for the rest of the sites"**
**p. 11 line 23, "This linear decomposition method a quantifies the contribution of different"**
All suggestions taken.

**p. 12 line 26, for clarity, if the following is correct I suggest revising this sentence to "The source water and leaf-water enrichment series forcing 18Op in MAIDENiso and ORCHIDEE are not significantly correlated over..." If this is not correct, sentence should be rewritten for better clarity.**
We meant that the dO18 of precipitation series were not correlated. This sentence was revised for clarity.

**p. 15 line 16, "in Fontainebleau and the other five sites across" line 17, "better simulates"**
Correction done.

**p. 17**
**line 9, "enables application of known mechanistic relationships between isotopes"**
**p. 20**
**line 24, "our findings suggest"**
**p. 23**
**line 5, author names should be "Anderegg, W. R. L." and "Williams, A.P."**

**p. 24**
**line 34, italicize "Pinus radiata"**
**p. 25**
**lines 21 and 22, "Fritts, H.C."**
**p. 26**
**lines 18-19, too many words in title are upper-case**
**p. 27**
**line 2, superscript "18"**
**line 23, italicize "Pinus radiata" line 32, superscript "13"**
**p. 28**
**line 31, subscript "2" line 33, superscript "18"**
**p. 29**
**line 5, superscript "13"**
All corrections done. We thank the reviewer for spotting these issues.

**Anonymous Referee #3**

**In this revised manuscript, the authors present a compelling case for using triple tree ring-proxies and their interactions as means to evaluate land surface model. The manuscript was substantially re-organized and the story is clear and reads well. The authors addressed most of the concerns raised in my previous review. This will be a relevant contribution for many communities in Biogeosciences, proposes an assessment of model parameters improvement, and demonstrate a tangible potential fo using tree ring data to evaluate land surface models.**
We thank this anonymous reviewer for her/his careful reviews of our manuscript and helpful suggestions, which improved the final version of the manuscript.

**Below I have minor comments for the authors to consider.**
**- The manuscript uses heavily acronyms (e.g. ITRDB), some of which were not spelled out and defined when first cited. I urge the authors to carefully check the text for acronym.**
We revised the acronyms and corrected the mistake of not spelling out ITRDB in its first occurrence in the introduction.

**- In their response, the authors argue that using the McCarroll and Loader 2004 atmospheric data has been the standard in the community (likely tree ring community). However, regardless of recent suggestion outlined in Belmecheri & Lavergne and your choice for not using the most updated data, note that the data used in McCarroll and Loader 2004 were recalibrated and are now the forcing data in CMIP6 efforts. I understand that ultimately the results will not change (interannual correlation) and the authors do not wish to re-run the analysis but it ought to be consistent in future work to use the most recent, updated, calibrated and vetted forcing data (similarly to model runs).**
This advice will be taken in future studies to keep up with the most updated standards.

**Detailed suggestion broken down by manuscript sections:**
**Introduction**
**- The following need references.**
**"a quasi-global network of eddy-covariance observations, Earth observations and forest inventories covering the last few decades"**
The following references were added in corresponding order: Baldcocchi (2019), Orth et al (2017), Bellassen et al. (2011)

**Methods:**
**- In the following, the fractionation parameters as well as the definition if the model parameters while summarized in McCarroll and Loader 2004, the latter in not the accurate reference for the model parameters and their values. Please cite the appropriate references.**
**"where a (4.4‰) is the kinetic discrimination associated with diffusion between free air and the stomatal cavity, b (27‰) is the fractionation during CO2 fixation by the Rubisco enzyme, ci is the leaf internal CO2 concentration simulated by ORCHIDEE and ca is the atmospheric CO2 concentration prescribed from measurements (McCarroll and Loader, 2004)."**
The references were revised and Farquhar et al. 1982 and its update Cernuzak et a., 2013 were used instead of McCarroll and Loader 2004.

**In the following "Six previously published tree-ring sites in northern and western Europe with simultaneous measurements of ring-width and..." one suggestion is to reword as: Six previously published tree-ring chronologies from sites in...**
It was revised to "Six previously published tree-ring datasets in northern and western Europe…"

**Discussion:**
**- "Disentangling the effect of isotopic signatures from source water and leaf evaporative enrichment revealed that the better performance of LPX-Bern to simulate _18 O variability was due to its higher sensitivity to leaf evaporative enrichment compared with ORCHIDEE (Fig. 3a), which is predominantly sensitive to source water..."**
**wouldn't this be dependent on site locations? where leaf water enrichment in drier sites will be a more important driver compared to more mesic sites where leaf water enrichment will not dominate the cellulose 18O. How does LPX-Bern parametrize the leaf water enrichment sensitivity or contribution. Is the latter variable per PFT/climate zone or just overall more important than the other 2 models?**
As shown in Fig .2, LPX-Bern systematically outperformed ORCHIDEE in our six study sites. Although the contributions of leaf water enrichment and source water will vary in nature according to moisture regimes, this result seems to indicate that at least at our boreal and temperate mesic sites leaf water enrichment seems to much more important than represented in ORCHIDEE. As explained in the paragraph following this sentence, both LPX-Bern and ORCHIDEE use the same representation for the Peclet effect but their parameterizations for L and mixing proportions differ. It is also indicated that for each model the parameterization for L is the same for all PFTS. We believe this is clearly addressed in the first paragraph of p17 but revised the sentence of line 23 to "while we used a fixed value of 8 mm for all PFTs obtained…" in order to be more explicit on that these parameters do not vary by PFT.

**- P17. Lines 25. This is a relevant suggestion for testing the L and péclet effect sensitivity in the model world. The uncertainties and sensitivity of these parameters hold also true for the empirical world when direct comparison of simulated and observed cellulose 18 O showed variability depending on (1) parameter values; but also (2) a model with or without the péclet effect. The péclet effect and effective path length are note measured but inferred and can vary seasonally or depending on aridity gradients as well. In sum, this not an uncertainty in the model parameters.**
We agree that the Peclet effect and L are highly uncertain and warrant careful estimation and sensitivity analysis. We encapsulate this issue in our closing remark in line 29 in p17 "Future studies should evaluate model sensitivity to these parameters and constrain them with tree-ring observations."

**- P17. Lines 30. This assessment is rather puzzling, it implies that 18O of precipitation is not a significant driver while it also determines the soil water? This warrant more clarification. The lack of correlation between the two models suggest that interannual variability is not important but the average value is? What would the correlation be if one precipitation (constant) value was used instead? This can show that**

**interannual variability or even seasonal change are not important but rather the integration period and average source water value.**

The dampening effect of source water mixing in the soil is something that has been observed in earlier modeling studies. We cite Danis et al., 2012. They evaluated the effect of different precipitation 18O forcing series (empirical vs model-based) and found that the interannual variability in source water signatures did not differ much because of soil water mixing. We believe we explain this clearly enough in p17 where we state "However, the dampening effect of soil water mixing reduces considerably the impact of the seasonal variability in $\delta 18O$ of precipitation on the more integrated $\delta 18O$sw signal (Danis et al., 2012)." Doing the test using climatological $\delta 18O$ of precipitation would help quantifying the effect of the integration period but is beyond the scope of this paper. This is certainly something that warrants further analysis, as we imply in our closing remark in that paragraph "A dedicated model intercomparison study for tree-ring $\delta 18O$ using the same meteorological and isotopic forcing will help to evaluate uncertainties and attribute differences to forcing data, parameters and model structure."